# Expanding the binding specificity for RNA recognition by a PUF domain

Wei Zhou[1,2,3,8], Daniel Melamed[1,4,5,8], Gabor Banyai[3,8], Cindy Meyer[3], Thomas Tuschl [3], Marvin Wickens [6], Junyue Cao [3✉] & Stanley Fields [1,7✉]

The ability to design a protein to bind specifically to a target RNA enables numerous applications, with the modular architecture of the PUF domain lending itself to new RNA-binding specificities. For each repeat of the Pumilio-1 PUF domain, we generate a library that contains the 8,000 possible combinations of amino acid substitutions at residues critical for RNA contact. We carry out yeast three-hybrid selections with each library against the RNA recognition sequence for Pumilio-1, with any possible base present at the position recognized by the randomized repeat. We use sequencing to score the binding of each variant, identifying many variants with highly repeat-specific interactions. From these data, we generate an RNA binding code specific to each repeat and base. We use this code to design PUF domains against 16 RNAs, and find that some of these domains recognize RNAs with two, three or four changes from the wild type sequence.

[1] Department of Genome Sciences, University of Washington, Seattle, Washington, USA. [2] Molecular and Cellular Biology Program, University of Washington, Seattle, Washington, USA. [3] The Rockefeller University, New York, NY, USA. [4] Department of Evolutionary and Environmental Biology, University of Haifa, Haifa, Israel. [5] Institute of Evolution, University of Haifa, Haifa, Israel. [6] Department of Biochemistry, University of Wisconsin-Madison, Madison, WI, USA. [7] Department of Medicine, University of Washington, Seattle, Washington, USA. [8] These authors contributed equally: Wei Zhou, Daniel Melamed, Gabor Banyai. ✉email: jcao@rockefeller.edu; fields@uw.edu

RNA targeting in living cells presents a unique opportunity to monitor and manipulate the diverse biological functions performed by RNA. One approach toward such targeting is via the engineering of protein-based RNA-binding domains to recognize an arbitrary RNA sequence. This engineering requires that the RNA-binding domain can be programmed for specific recognition with limited off-target effects. However, decoding the RNA-binding specificity of most types of RNA-binding domains is challenging, as these domains can associate with RNA via complex networks of interactions. For example, the most abundant RNA-binding domain in vertebrates, the RRM domain[1,2], has two conserved ribonucleoprotein consensus sequences (designated RNP1 and RNP2). More than 30 structures of RRMs in complex with RNA are available, making it the most intensively studied class of RNA-binding domain[2-5]. However, it is still difficult to predict recognition specificity solely from the amino acid sequence of an RRM domain. The RRM domain can form distinct sub-states at the RNA-binding surface to allow for high or low-affinity binding, and inter-domain interactions and the linker region between multiple RRMs within a protein also contribute to RNA recognition[1,2].

Another major class of RNA-binding domains—the pentatricopeptide repeat (PPR)—uses a modular structure arranged in tandem repeats, with each repeat binding primarily to a single RNA base. PPR proteins have two amino acids that provide specificity and make this domain highly useful for targeting RNA by design, although binding sites for certain naturally occurring variants are unclear[6-10]. PPR proteins can be insoluble in heterologous systems, which has hampered biochemical characterization of specificity and hence design[9]. PPR redesign has identified proteins that bind to a new RNA target[11,12], but complications exist. For example, a comprehensive analysis of the sequence specificity of the native protein PPR10 showed that four of the 17 nucleotides in a target RNA are specified in a manner that cannot be explained by the current PPR code[13].

The Cas13 family and CRISPR RNA technology provide a powerful approach to manipulate RNAs in vivo. The strategy achieves specificity through guiding RNA interactions with a targeted RNA sequence. The approach has been applied to detect, modify, cleave, image, edit, and manipulate RNA processing[14-16] and has been applied in cells ranging from bacteria to mammalian cells[17-21]. Reliance on RNA–RNA interactions provides a strong underpinning for rational design. Challenges include the requirement to deliver multiple components, the large size of the protein, and off-target effects[20].

PUF proteins are involved in regulating eukaryotic processes that include embryogenesis and development[22-25]. They contain a conserved RNA-binding domain, known as the Pumilio homology domain or PUF domain, that is generally composed of eight 36-amino-acid repeats (Fig. 1A)[26,27]. Each repeat displays three amino acid residues, called the tripartite recognition motif (TRM) combination[28], on the concave surface of the protein. A target RNA sequence of eight bases is bound as an extended strand to the concave surface. X-ray structural analysis of the complex indicates that the recognition is highly modular, with each repeat binding to a single RNA base[26,27]. Residues at positions 12 and 16 in each repeat directly interact with a Watson-Crick edge of a base, whereas the residue at position 13 is involved in a stacking interaction with the base[26,27].

The identities of the residues in a TRM combination (here, in XXX format of three amino acids, left to right indicates positions 12, 13, and 16) play a key role in RNA-binding specificity[29-31]. For example, at the 12 and 16 positions, respectively, cysteine and glutamine bind adenine; asparagine and glutamine bind uracil; and serine and glutamate bind guanine. No TRM combinations in

natural PUF proteins have been found to specifically bind cytosine.

PUF domains have been engineered to design new sequence specificities and repurposed to control multiple steps in RNA biology. Different TRM combinations can shift specificity from one nucleotide to another, while others broaden rather than switch specificity. For example, Cheong and Hall[30] mutated repeats 1, 3, and 7 of the human Pumilio-1 domain and changed specificities to guanine and uracil. Ozawa et al.[32] mutated multiple repeats of the Pumilio-1 domain to target sequences of a mitochondrial mRNA. Efforts to identify TRM combinations that recognize cytosine imply that amino acid sequence context can influence TRM specificity. Residues of Pumilio-1 that are capable of specifying cytosine have been identified, and are applicable in different positions in the RNA[33,34]. A study using C. elegans FBF-2 determined the specificities of 25 natural and engineered PUF variants in repeat 7 of this protein and proposed a code for guanine, uracil, and adenine recognition, but did not detect a cytosine-specific code. The differences in these studies prompted the suggestion that the context in which the TRM appears can influence specificity[28]. Multiple studies of PUF redesign and specificity[24,25,34,35] support this conclusion, and suggest that any PUF code may not be generic for all repeat locations[34]. Furthermore, PUF proteins can have an elongated RNA-binding surface that recognizes nine bases, with the central bases flipping away from this surface[36], which may also affect PUF design prediction.

We sought to further understand the basis of PUF specificity using a broad selection strategy in Saccharomyces cerevisiae. We explored the binding specificity and affinity of TRM combinations that can specify any of the four RNA bases in any of the eight positions. Beginning with the human Pumilio-1 RNA-binding domain and its eight-nucleotide binding site, we tested libraries of ~8000 possible variants of TRM combinations in each repeat against sets of four RNAs altered at the cognate RNA position. By combining the yeast three-hybrid method with next-generation sequencing, we scored the binding activities of all of these PUF variants for each of the four possible RNA bases. We identified many variants with highly specific interactions, suggesting that each repeat may need to be optimized for maximal binding to an arbitrary RNA sequence.

## Results

**A high-throughput assay to globally identify PUF variants with new specificities.** To quantify the interaction between a PUF domain and its RNA target, we applied the yeast three-hybrid assay[37] (Fig. 1B). In this assay, the interaction of the PUF domain with an RNA leads to the activation of the yeast reporter gene HIS3, such that the cells survive in selection media without histidine and containing 3-amino-1,2,4-triazole (3-AT), a competitive inhibitor of His3. In order to facilitate the identification by next-generation sequencing of both a protein variant and its RNA target, we encoded both components on a single plasmid. This "all-in-one" plasmid includes both a protein module encoding the human Pumilio-1 PUF domain fused to the Gal4 activation domain (PUF-AD) and an RNA module encoding the Pumilio-1 RNA recognition sequence, the nanos response element (NRE)[38], fused to the MS2 coat protein binding sequence (Fig. S1A). We chose the NRE element UGUAAAUA as the starting recognition sequence rather than a sequence with U or C at position 5 (ref. [26]), as UGUAAAUA is the most common Pumilio-1 binding motif based on a comparative analysis of mRNA targets for the human PUF family proteins[39], a structure exists for human Pumilio-1 bound to this sequence[30], and A was one of three

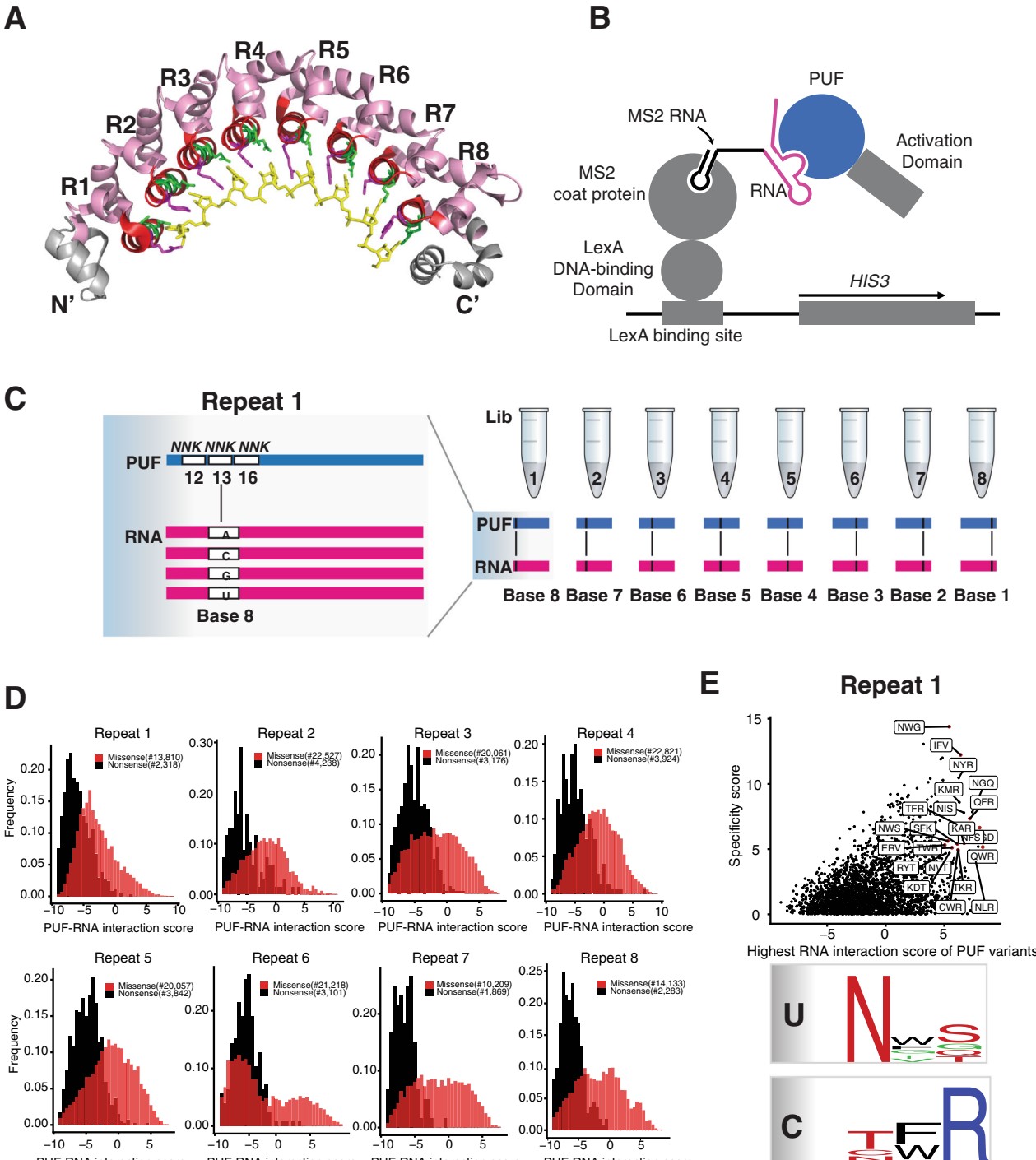

**Fig. 1 A high throughput yeast three-hybrid assay to identify PUF variants. A** Crystal structure of the human Pumilio-1 PUF domain in complex with NRE10 RNA (PDB ID 1M8Y)[26]. Helices that carry functional RNA-binding residues (TRM residues) are colored in green and purple. TRM residues 12 and 16 are in green and position 13 is in purple. The RNA bases of the NRE10 recognition sequence are colored in yellow. **B** The yeast three-hybrid system was adapted for deep mutational scanning of the Pumilio-1 PUF domain TRM residues. Binding of the Pumilio-1 PUF domain to its cognate RNA sequence leads to the formation of a functional transcription factor that induces the expression of the reporter gene, *HIS3*. As a result, yeast cells that carry functional PUF-RNA interactions proliferate in media lacking histidine, while yeast cells that carry non-functional PUF-RNA interactions will be eliminated. **C** Workflow to analyze all possible TRM combinations (~8000) against the four possible RNA bases for each PUF repeat through deep mutational scanning. The left panel is shown as randomization of Repeat 1 and Base 8. **D** The frequency distribution of nonsense and missense PUF variants for all repeat locations. The *X*-axis is a measure of PUF-RNA interaction score. Black indicates nonsense variants and red indicates missense variants. **E** A plot showing the RNA interaction score and specificity score of each PUF variant in repeat 1. The *X*-axis indicates the highest RNA interaction score of each PUF variant. The *Y*-axis indicates the specificity score for each PUF variant. Each dot indicates one PUF variant and the red dot highlights those PUF variants with interaction score >5 and specificity score >4. The lower panel summarizes the base-specific recognition pattern for uracil and cytosine through sequence logos.

preferred bases at that position when Pumilio-1 binding was tested in the yeast three-hybrid assay (Fig. S2).

The use of an "all-in-one" plasmid also simplified plasmid recovery from yeast containing a library of PUF domain variants. We performed six tests of the plasmid with combinations of RNA and PUF domain in the yeast strain *YBZ-1*, which constitutively expresses the LexA–MS2 coat protein fusion and carries the *HIS3* reporter gene under the control of multiple LexA operators. We found that the all-in-one plasmid performed similarly to the two plasmids of the original yeast three-hybrid system (Fig. S1B; 1 vs. 2), and that high copy and low copy versions of the plasmid also worked similarly (Fig. S1B; 2 vs. 3). In the last three tests, we swapped the TRM combination in two repeats, or we swapped two RNA bases, or both. Either swap should eliminate RNA–PUF domain binding, which was what we observed (Fig. S1B; 4, 5, and 6).

We tested the PUF domain against RNA targets that sequentially contained each of the four RNA bases, with the other seven positions being the wild-type base. The base-specific binding pattern for each PUF repeat could be recapitulated in this system (Fig. S2). Only base 5, an adenine in the Pumilio-1 target sequence, showed a broader specificity, generating a three-hybrid signal when either adenine, cytosine or uracil was present. This broader specificity has been observed previously[39–41]. Overall, these results confirmed that the yeast three-hybrid assay can be used to analyze a PUF domain binding to its RNA target.

To elucidate the RNA-binding preferences of a large number of PUF variants in a single culture, we combined the yeast three-hybrid system with next-generation sequencing. For each of the eight repeats of the PUF domain, we generated a library of all possible TRM domains. Each library was encoded on a plasmid that also carried the Pumilio-1 target RNA sequence with any of the four possible RNA bases present at the cognate position of the 8-base binding site. Thus, each of eight separate three-hybrid selections tested a single TRM library of the PUF domain against a target RNA sequence with a single base varied. To identify protein-RNA interactions by single short reads of Illumina sequencing, we designed the TRM libraries to carry synonymous changes in codons adjacent to the randomized TRM codons. For each TRM library we used four sets of synonymous changes, which informed the identity of the cognate RNA base that was varied (see "Methods"); the synonymous changes were likely to have a negligible effect on protein function.

The PUF domain variants were designed to contain the 8000 possible combinations of amino acid substitutions at residues 12, 13 and 16 through NNK libraries at each position (N = A/C/G/T and K = G/T, Fig. 1C). We selected for the ability of each repeat to interact with RNA by carrying out the histidine selection on plates with SC-Leu-His + 0.5 mM 3-AT media, a 3-AT concentration chosen based on the pilot selection (Fig. S1B). We retrieved the library from both input and post-selection pools, and determined the frequency of each variant in both pools by high-throughput sequencing. The log2 change in the frequency from input to selection pool serves as a measure of binding activity for each PUF variant, designated as a PUF domain–RNA interaction score in this assay.

Based on enrichments in the post-selection pool, we scored the RNA-binding activities of 169,587 PUF domain variants. This dataset contains 24,751 nonsense variants and 144,836 missense variants from the eight repeats. The interaction score distribution of all variants revealed that, in general, nonsense PUF variants were deleterious for interaction with any RNA sequence, and missense PUF variants were present as a bimodal distribution (Fig. 1D). Some nonsense variants had scores that indicated they were enriched, which may result from experimental noise, as routinely seen in other deep mutational scanning experiments[42];

the nonsense variants with these enrichment scores had significantly lower input reads than other nonsense variants. The use of these scores allowed us to calculate a false positive rate for loss-of-function missense mutations. We found that 1.4% (45/3193) of nonsense variants had an enrichment score >0, providing an estimate of the fraction of the loss-of-function missense variants that were also false positives.

The PUF domain–RNA interaction score for each PUF variant showed a high degree of overlap between two experimental replicates (Fig. S3; Pearson correlation coefficient ranged from $R = 0.952–0.982$). We assigned a specificity score for each PUF variant as the difference between its highest and second-highest interaction score. Using a threshold of interaction score >5, and a specificity score >4, we identified many PUF variants with highly specific interactions (Figure S4; the number of enriched PUF variants ranged from 5 to 79 across the eight repeats). For example, in repeat 1 (Fig. 1E), we found nine PUF variants specific for uracil (e.g., NWS, NFS), 11 PUF variants specific for cytosine (e.g., TFR, QFR), one PUF variant specific for guanine (SGD), and one PUF variant specific for adenine (IFV). For uracil recognition in repeat 1, we found that asparagine was the most preferred amino acid in position 12, and a polar uncharged amino acid such as glutamine, serine or threonine was the most preferred in position 16. Position 13 was enriched in the aromatic amino acids tryptophan and phenylalanine. While this pattern differs slightly from the optimal TRMs for uracil recognition (NXQ with X denoting T/H/F/Y)[43], it recapitulates the general trend. Similarly, for cytosine recognition in repeat 1, arginine was the most preferred amino acid in position 16 and a polar uncharged amino acid (e.g., threonine, glutamine, asparagine) was preferred in position 12[33,34].

**Targeted screening of candidate PUF variants.** Due to the large size of the libraries of randomized PUF variants, for many TRM variants, the initial yeast three-hybrid screen did not comprehensively recover a binding activity score against all four RNA bases and across all eight repeats. We thus conducted a targeted three-hybrid screen of promising candidate PUF variants. Using a threshold of interaction score >5, as well as specificity score >4, we chose for targeted oligonucleotide synthesis about 250 candidate PUF variants (along with negative controls of nonsense and missense variants) for each repeat (Supplementary Data 1; the number of variants ranged between 181 and 299) (Fig. 2A). We cloned each oligonucleotide pool into one of the eight PUF repeats to comprehensively survey the interaction of the candidate variants. For this experiment, we carried out 32 separate three-hybrid selections, consisting of the ~250 variants of a PUF repeat against one of four RNAs with a single cognate base varied. For each selection, to compare the binding of the wild-type Pumilio-1 domain across the four RNA bases, we spiked in the wild-type domain for normalization. We again collected plasmids from both input and post-selection pools and measured the change in frequency of each PUF variant by high-throughput sequencing.

For each repeat, we recovered between 64% and 95% of the synthesized PUF variants in the input pool (Fig. 2B), with each variant having a frequency centered on 0.1% (Fig. 2C). PUF variants were assigned an interaction score based on their enrichment in the post-selection pool. The distribution of interaction scores for all nonsense variants indicates that they were mostly deleterious, with an interaction score < −5. Consistent with the initial screen, targeted missense variants were enriched in the post-selection pool (Fig. 2D).

Based on the interaction scores against the four RNA bases, for each repeat, we clustered promising PUF variants with interaction scores >0. Of these, we identified variants with

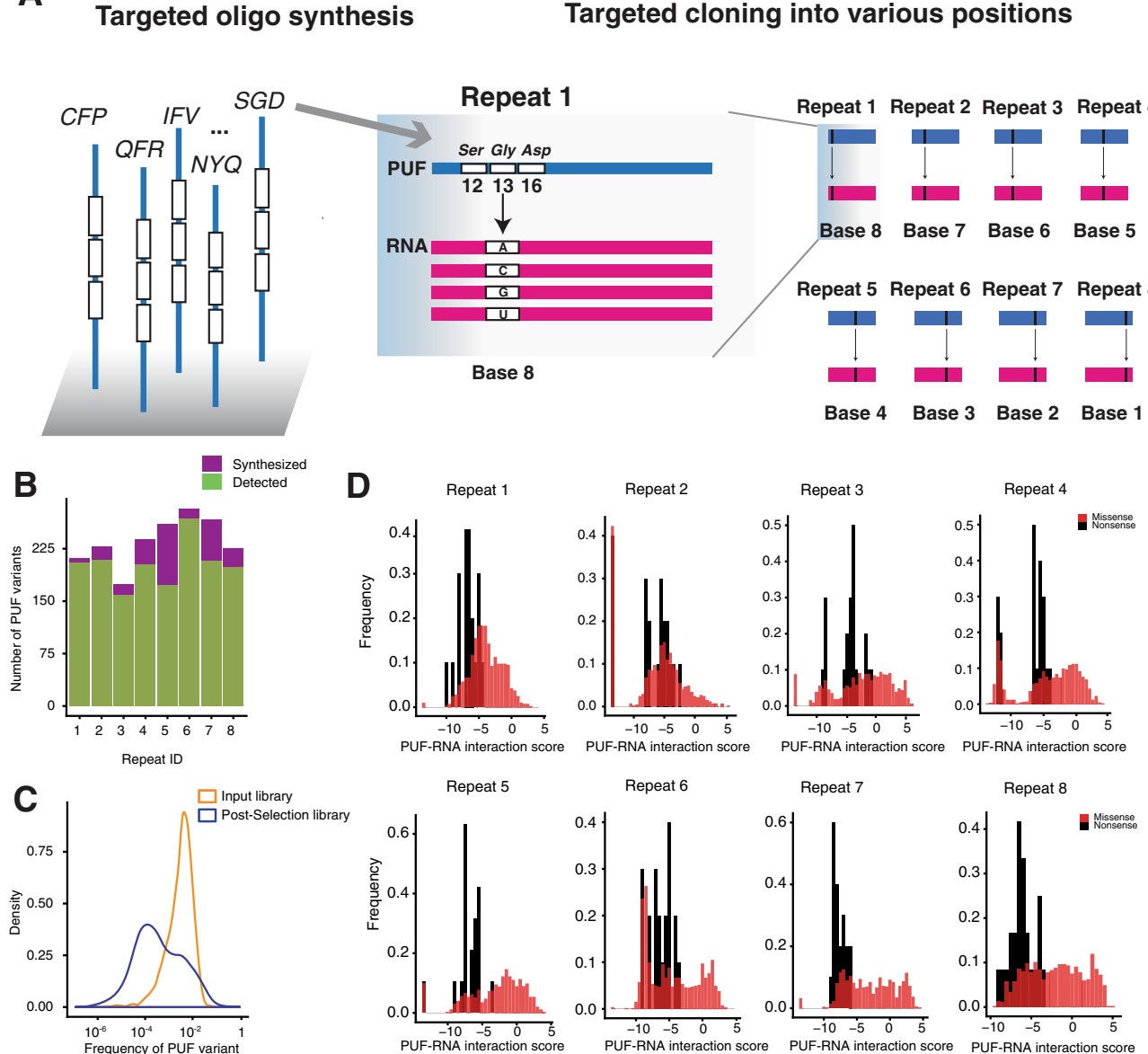

**Fig. 2 Targeted screening of candidate PUF variants. A** Workflow for the targeted screening experiment. The left panel shows examples of oligos designed for synthesis; the middle panel shows one example incorporated into the library repeat 1; the right panel shows a schematic of incorporation at all repeat locations. **B** A bar plot showing the fraction of targeted PUF variants recovered in each repeat. Purple indicates the number of synthesized PUF variants and green indicates the number of recovered PUF variants in each repeat. **C** A density plot showing the frequency of PUF variants in input and selected pool. Yellow indicates input library and blue indicates post-selection library. **D** The frequency distribution of nonsense and missense PUF variants for all repeat locations. The *X*-axis is a measure of PUF-RNA interaction score. Black indicates nonsense variants and red indicates missense variants.

highly base-specific interactions for each of the eight repeats, and generated sequence logos for those PUF variants that had specificity scores >4 (Fig. 3). For clusters with more than ten variants, we subclustered the variants based on the properties of the amino acids across the three positions (e.g., positively or negatively charged or neutral) and generated separate sequence logos for each subcluster to summarize the base-specific recognition patterns. The top TRM combinations for each PUF repeat and each base in the cognate position are shown in Fig. 4. Comparing the specificity of each PUF variant across the eight repeats, we found that many base-specific recognition codes are not generic for all repeat locations, as previously reported[34].

For G-specific binding, SNE is the natural code in repeat 7 (ref. [43]). We found that this code specified guanine in repeat 3

as well (the interaction score for the other three bases was <40% of the score for G). If, instead, tryptophan was present in position 13 (SWD), G-specific recognition could be achieved in repeats 6 and 7. Moreover, if position 13 was glycine (SGD), G-specific recognition could be achieved in repeat 1, with SNE and SWD not found (Fig. 5). These results suggest that the combination of serine and a negatively charged amino acid (aspartate or glutamate) in positions 12 and 16, respectively, was a trend for G-specific binding across the majority of the repeats, with an aromatic amino acid (W/Y/F) in position 13 also affecting recognition specificity. In addition, in some repeats such as repeat 3 and repeat 6, a combination of threonine in position 12 and a negatively charged amino in position 16 (e.g., THE, Fig. 5) achieved guanine-specific binding.

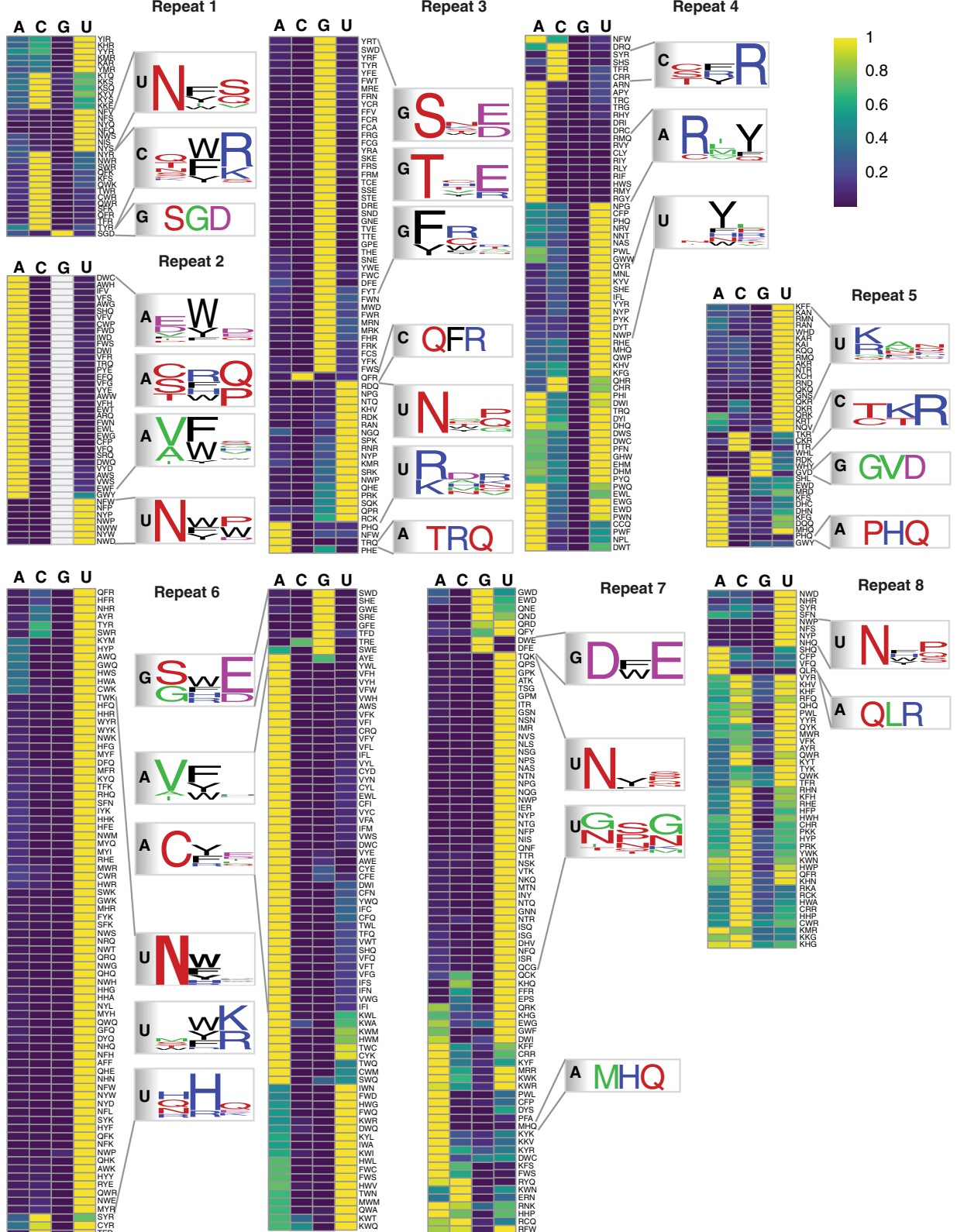

**Fig. 3 Heatmaps showing unsupervised clustering of PUF variants with interaction score >5 for repeats 1–8.** The color intensity represents the relative interaction score normalized by the maximal value for each row. Yellow indicates a high interaction score and blue indicates a low interaction score. Sequence logos summarizing the base-specific recognition patterns are shown nearby the heatmap for each repeat.

| | A | C | G | U |
|---|---|---|---|---|
| R1 | *SRQ | QFR<br>TFR<br>SFK | SGD | NYQ<br>NFQ |
| R2 | VFQ<br>CFP<br>SRQ | --- | --- | *NYQ<br>NFW<br>NFP |
| R3 | TRQ<br>*CRQ | QFR | SNE<br>THE<br>TTE | NTQ<br>NPG |
| R4 | RGY<br>RMY<br>RLY | SYR<br>TFR<br>CRR | --- | NWP<br>NYP |
| R5 | PHQ<br>*CRQ | TTR<br>TKR<br>CKR | GVD | RAN<br>QKQ<br>GNS |
| R6 | VFH<br>VFW<br>VWH | --- | SWD<br>SHE<br>SRE | *NYQ<br>TWK<br>HHR |
| R7 | MHQ | KWN<br>ERN | *SNE<br>DWE<br>DFE | NPG<br>NPS<br>NSG |
| R8 | QLR | --- | --- | *NYQ<br>NWP<br>NYP |

**Fig. 4 Summary of the best TRM combinations for each PUF repeat.** These TRM combinations follow one of three criteria: (1) used in the wild-type Pumilio-1 PUF domain, indicated by * in the figure; (2) highly specific in both the random screen and targeted screen; or (3) best represent the pattern of the sequence logo in Fig. 3. The rows indicate each repeat position of the Pumilio-1 PUF domain, and the columns indicate the four RNA bases at the cognate positions. Red, TRM combinations used in the PUF domain designs; ---no highly base-specific TRM combination detected.

For A-specific binding, the natural base-specific combinations (C/S)RQ[28] were recapitulated as SRQ across many repeats (Fig. 5). However, in repeats 1, 2, 6, and 8, the combination of a valine or cysteine in position 12 and phenylalanine in position 13 (CFP, VFQ) was an alternative way to achieve A-specific binding (Fig. 5). While NHQ is a natural TRM combination that specifies uracil, replacing asparagine by proline in position 12 (PHQ) resulted in adenine specificity for repeat 3, 5, and 7 (Fig. 5). These results further indicate that base-specific combinations other than canonical codes can be identified.

For U-specific binding, the natural TRM combination NYQ was found highly specific in repeat 1, 3, and 7 (Fig. 5). Asparagine was preferred in position 12 (NHQ, NWP) across the majority of repeats (Fig. 5). For the middle repeats, a positively charged amino acid was more preferred than a polar residue in position 12 or 16 (e.g., RAN; Fig. 5). For position 13, aromatic amino acids such as phenylalanine or tyrosine were preferred. Even for canonical base-specific combinations, each repeat had its own preferences. For example, repeat 1 preferred NHQ rather than NWP, while the opposite was the case for repeat 2 (Fig. 5).

For C-specific binding, a polar, uncharged amino acid (e.g., glutamine or threonine) in position 12 and a positively charged amino acid (e.g., arginine) in position 16 (TFR, QFR, QWR) were the preferred combinations (Fig. 5). However, this preference was not uniform across the eight repeats. High specificity for cytosine was found only in the more N-terminal repeats, such as repeat 1 or 3, and was markedly reduced in more C-terminal repeats (Fig. 5). The same pattern was seen for the previously identified C-specific codes (e.g., SYR)[33,34] as well, which potentially explains why C-specific combinations identified from repeat 6 in human Pumilio-1 did not confer this recognition to repeat 7 of C. elegans FBF-2[34,43].

Many combinations showed non-specific binding. For example, repeat 1 combinations with positively charged amino acids in position 12 and 16, repeat 3 combinations with negatively charged amino acids in position 12, and repeat 6 combinations with an aromatic in position 13 and an arginine in position 16 bound to more than a single base, with some of these combinations previously characterized as capable of non-specific RNA recognition[27].

**Binding of designed PUF domains against target RNA sequences.** Given this new set of TRM combinations specific for each of the eight Pumilio-1 PUF repeats, we sought to determine their utility to bind in combination to RNA sequences possessing multiple changes to the UGUAAAUA wild-type binding site. Toward this end, we generated 16 8-base target RNA sequences that differ from the wild-type site by either one base (one target), two bases (two targets), three bases (eight targets), four bases (three targets), five bases (one target) or six bases (one target) (Fig. 6A). RNA targets with successively larger numbers of changes were generally devised to include changes put into the less heavily substituted targets. We tested the effects of RNA changes that require recognition by N-terminal and C-terminal repeats; single and consecutive (up to four base) changes; and changes that resulted in substitutions to A, C, G, and U.

To assay binding to these targets, we generated an array of 1900 RNA elements (Supplementary Data 2). The array contained 20 copies of the wild-type sequence and 55 copies of each of the 16 target sequences (totalling 900), with the remaining 1000 sequences containing different percentages of the four bases in each location of the 8-base RNA sequences. For each base, we calculated these percentages to ensure that the oligo pool included related off-target sequences (see "Methods"). To synthesize the PUF domains, we chose TRM combinations identified as specific to each repeat, based on the results from both screens. For example, we chose TRM combinations SWD in repeat 6 and THE

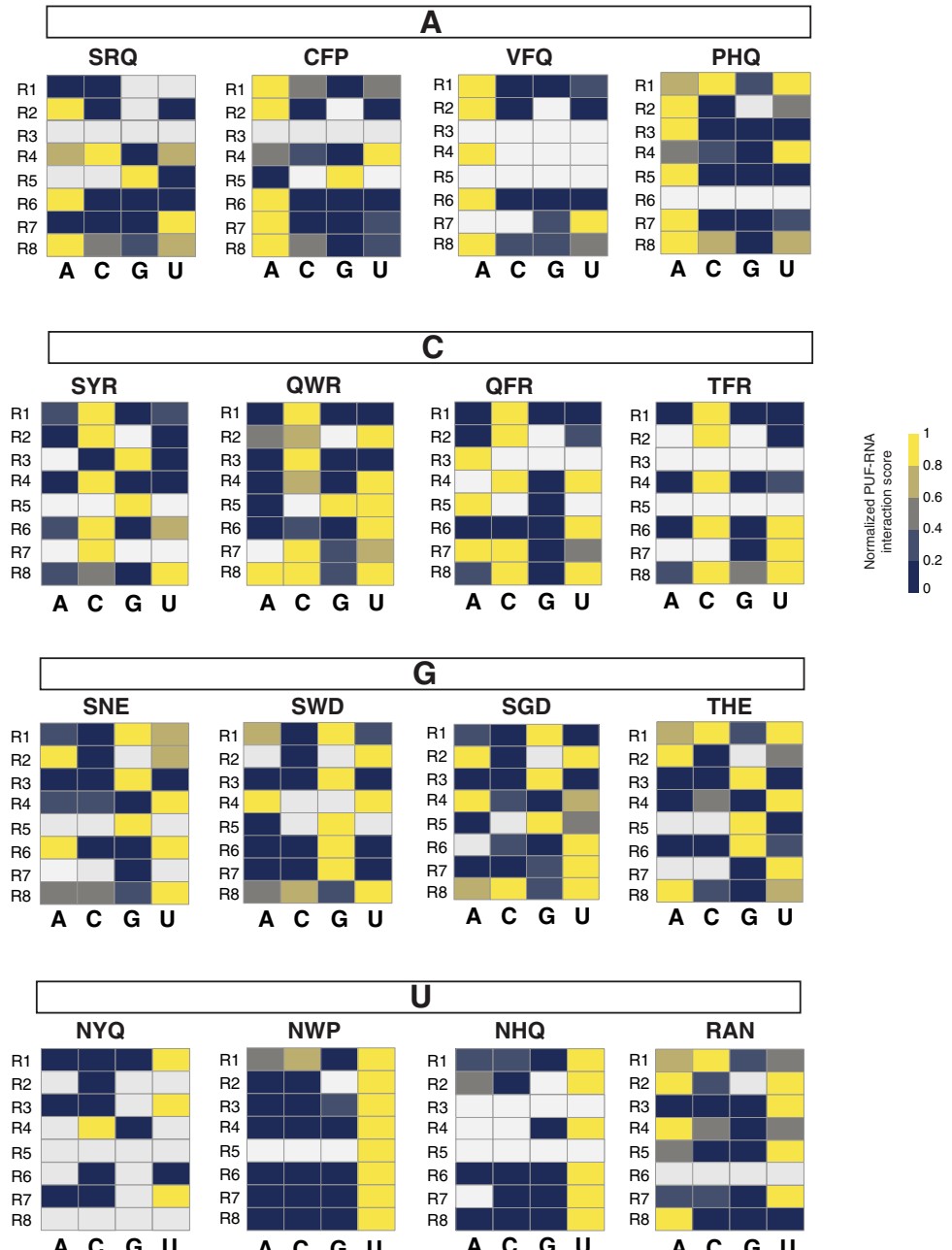

**Fig. 5 Comparison of base-specific recognition patterns across repeats.** The four TRMs shown for each base are representative of TRMs with different behavior in different repeats. The heatmap shows the normalized interaction score for each base. The color intensity represents the relative interaction score normalized by the maximal value for each row. Yellow, high interaction score; dark blue, low interaction score; white, missing data. The box above each heatmap panel indicates the base that the TRM combinations prefer.

in repeat 3, as they contain the canonical guanine-specific recognition pattern of polar, uncharged amino acids in position 12 and a negatively charged amino acid in position 16, but they differ from the exact combinations in natural proteins. We chose other TRM combinations that were highly specific, but differed from canonical recognition patterns. For example, VFQ was the best combination for A-specific recognition in repeats 1, 2, 4, and 6; here we chose it for repeat 2. For cytosine recognition, we chose the most specific combinations QFR in repeat 1 and TFR in repeat 4.

We carried out 16 three-hybrid selections in duplicate corresponding to the 16 designed PUF domains, and calculated interaction scores for each RNA found in the input and selection pools. The sequence UCCGACUA was highly enriched in many of the selections independent of the TRM combinations that were substituted, suggesting that it resulted in reporter gene activation not due to a three-hybrid interaction. We thus removed this sequence and plotted the interaction scores of the remaining RNA sequences (Fig. 6B), with specific points labeled to show the target RNA, the wild-type RNA, RNAs differing from the target by one or two bases, and other RNAs that scored high for interaction.

For the three double PUF variants (including design 1, which has two TRM combinations changed but only one base changed in the target RNA), the target sequence was the most enriched RNA sequence (Fig. 6B, designs 1–3). For example, when repeat 3 was replaced with THE and repeat 4 with TFR, the target

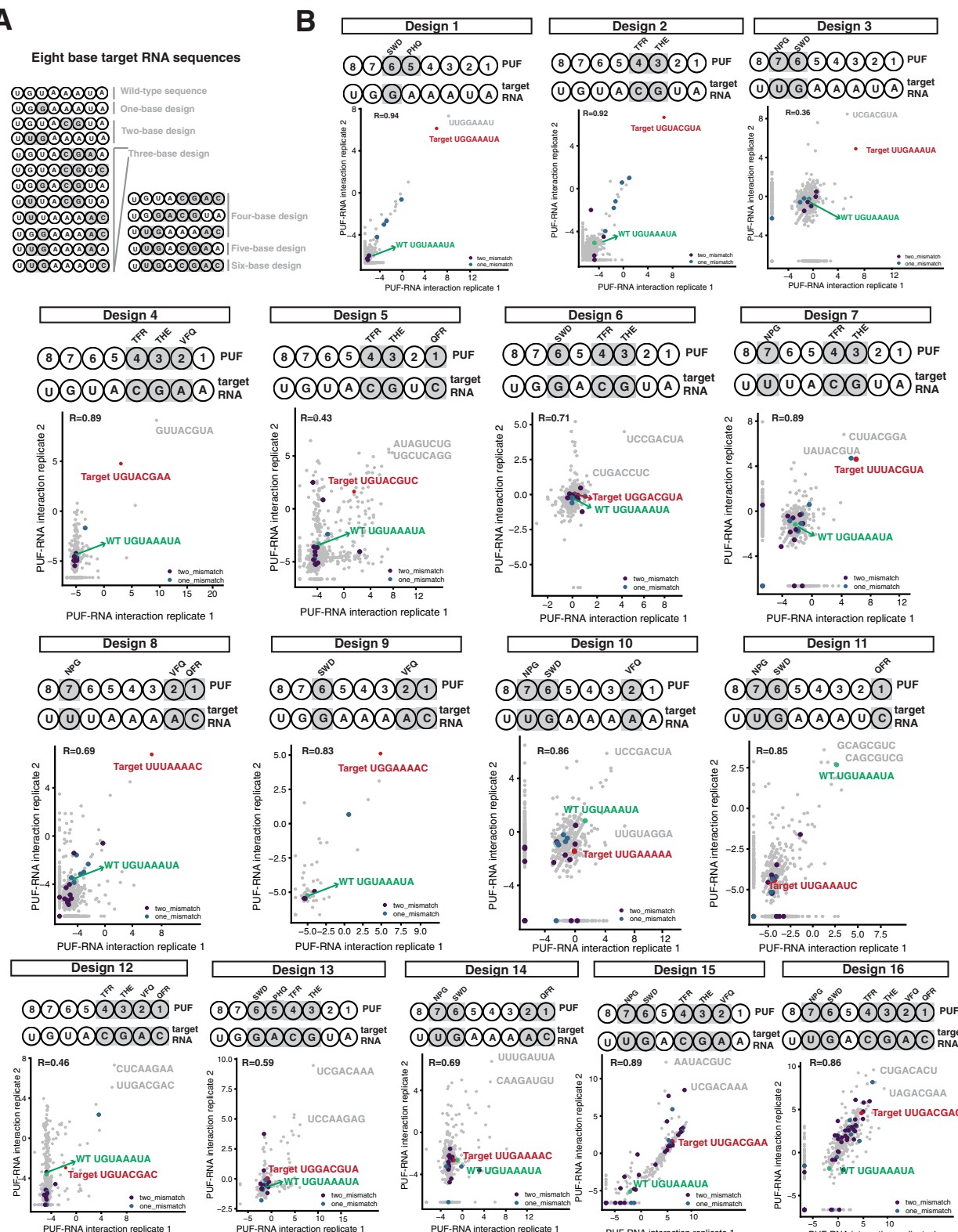

**Fig. 6 RNA sequences that bound to designed PUF domains. A** Target RNA designs, showing the substituted bases and designed TRM combinations in gray. **B** Sixteen three-hybrid selections were carried out in duplicate and interaction scores for each RNA in both replicates were plotted. The upper panel of each experiment indicates the locations and substitutions made in the PUF domain and target RNA. The lower panel is a correlation plot between the two replicates, with each dot an RNA sequence. Green indicates the RNA sequence recognized by the wild-type PUF domain; red indicates the target RNA sequence; gray indicates the top two enriched sequences based on the averaged interaction score from the two replicates, if these interaction scores are higher than the target RNA sequence. The X and Y axes indicate two replicates of the experiment. The sequence UCCGACUA was highly enriched in some of the replicates and was removed, and then the interaction scores of the remaining RNA sequences were plotted, which resulted in the different scales of the X and Y axes.

UGUACGUA was the most enriched RNA in both replicates (Fig. 6B, design 2). For the triple PUF variants (Fig. 6B, designs 4–11), the target sequences were enriched in half (four of eight) of the designs. For example, when repeats 1, 2, and 7 were replaced with QFR, VFQ, and NPG, respectively, the most enriched RNA sequence in both replicates was the target sequence UUUAAAAC (Fig. 6B, design 8). However, when repeats 1, 6 and 7 were replaced in a triple variant with QFR, SWD, and NPG, respectively, the target sequence UUGAAAUC had a low interaction score (Fig. 6B, design 11). For the quadruple PUF variants, none of them identified their target sequence as highly enriched (Fig. 6B, designs 12–14). Similarly, for the pentuple and sextuple variants, their targets were not among the top enriched RNA sequences (Fig. 6B, designs 15–16). However, in some cases, these highly mutated targets had higher interaction scores compared to the wild-type or many of the other RNA sequences (for example, Fig. 6B, designs 15–16).

In some cases, highly enriched RNA sequences that were not the targets matched part of the target sequence in a pattern that suggests the substituted repeats of the designed PUF domain were binding as designed. For example, in a triple variant with repeats 3, 4, and 7 replaced (Fig. 6B, design 7), the most enriched sequence matches six of eight bases of the target, including the three bases that were changed. For a quadruple variant with the TRM combinations substituted in repeats 1–4 (Fig. 6B, design 12), one of the most enriched RNA sequences was UUGACGAC. This sequence includes the four bases CGAC, which are the target sequence for the combination of four substitutions in repeats 1 to 4. However, the 5′-most four bases match only two of four bases of the target. Thus, this PUF design recognized all the substitutions in the RNA but no longer bound to all the remaining wild-type bases. For the sextuple variant (in repeats 1, 2, 3, 4, 6, and 7; Fig. 6B, design 16), the most enriched RNA sequence (UAGACGAA) includes five consecutive bases, GACGA, that match repeats 2, 3, 4, 5, and 6, corresponding to four of the substitutions in the RNA.

To determine whether flanking RNA bases beyond the 8-mer core provoked a register shift along a repeat that influenced the binding of the designed PUF domains, we compared the enrichment score of the target RNA to RNAs containing possible mismatched bases (Figure S5). For example, for any design, if a 1-base 5' shift occurred in recognition, then the enrichment score of the designed 8-mer target would be similar to the three 8-mers that have the same seven 5' bases and a different final base than the target in position 8; if a 2-base shift occurred, then the enrichment score of the designed 8-mer target would be similar to the nine 8-mers that have the same six 5' bases and a different final base than the target in position 7 or 8. Similar considerations would apply at the other end of the 8-mer if 3′ shifts occurred. We plotted the enrichment scores of these alternative 8-mers and found no evidence that shifting occurred for designs that bound to their target sequences; shifting may have occurred for some designs, such as designs 13, 15, and 16, that did not bind to their target sequences (Fig. S5).

Results from these targeted RNAs and cognate substitutions in the PUF domain designs suggest two features that may hold more generally. First, N-terminal repeats (1–4) appeared to tolerate combinatorial variation better than C-terminal repeats (5–8). For example, target RNA sequences were highly enriched in several triple variants with two N-terminal substitutions, whereas they were not in triple variants with two C-terminal substitutions (Fig. 6B, designs 4–11). This finding is consistent with reports that UGUA, the cognate sequence for the four C-terminal PUF repeats, is a conserved binding motif for PUF domains from different species[44,45]. Conservation of the UGUA sequence may limit the ability of the C-terminal repeats to recognize alternative bases.

Second, these data suggest that N-terminal repeats engage in crosstalk with C-terminal repeats. For double PUF variants with substitutions only in N-terminal repeats (e.g., 3 and 4; Fig. 6B, design 2) or only in C-terminal repeats (e.g., 6 and 7; Fig. 6B, design 3), the most enriched sequences were the target RNA sequences. However, the addition of another substitution on the other side of the PUF domain resulted in triple variants that did not bind to their target RNA sequences (e.g., substitution in repeat 6 added to substitutions in 3 and 4 (Fig. 6B, design 6), or substitution in repeat 1 added to those in repeats 6 and 7 (Fig. 6B, design 11)). Similarly, for quadruple variants, consecutive substitutions at a single terminus (e.g., repeats 1, 2, 3, and 4; Fig. 6B, designs 12) functioned better than separate pairs of substitutions at both termini (e.g., repeats 1 2, 6, and 7; Fig. 6B, design 14). Mutations present at both termini may inhibit folding of the PUF domain.

**In vitro binding of purified variant PUF domains.** We sought to use electrophoretic mobility shift assays with purified PUF variants to quantify their binding properties and compare the results to the yeast three-hybrid results. *HIS3* activity in the yeast assay correlates with biochemically measured protein-RNA affinity, but relatively small changes in Kd can cause substantial differences in 3-AT resistance[46]. We purified GST-fusions of PUF domains from *E. coli* using glutathione chromatography and removed the GST domain by protease cleavage. Initially, we examined the binding of our starting constructs, the wild-type Pumilio-1 PUF domain binding to the wild-type nanos response element UGUAAAUA (Fig. 7A, upper panel). The Kd estimated for this pair, based on half-maximal binding, was ~80 nM (95% CI 54–125 nM). This value is considerably higher than other estimates of Pumilio-1 binding, of around 1 nM[41], and may reflect loss of activity during purification. Nonetheless, the assay using the wild-type protein represents a baseline for comparison to PUF domains with variant TRM combinations. The wild-type domain showed no binding to either of two RNAs with changes to two bases.

We generated and purified a PUF variant (designated QFR/VFQ) with QFR (specific to C) in repeat 1 and VFQ (specific to A) in repeat 2. QFR is a newly identified TRM code for binding to C, with the QFR-C pair highly enriched and specific in our screens with TRM mutations in repeats 1 and 3; similarly, the VFQ-A pair was enriched and specific for repeat 2 (Figs. 3, 4). We cloned the QFR/VFQ variant and tested its binding via a yeast three-hybrid spotting assay, finding that it bound to its target in this assay similarly to the wild-type PUF domain binding to the wild-type RNA sequence (Fig. 7B). In the electrophoretic mobility shift assay, the QFR/VFQ variant bound to its target RNA beginning at the 15.6 nM concentration, with an estimated Kd of ~110 nM (95% CI 70–177 nM). While this binding is somewhat weaker compared to the value we determined for the pair of the wild-type PUF domain with the wild-type RNA sequence, it was completely sequence specific: the wild-type PUF domain did not bind to the variant-specific target RNA, and the QFR/VFQ PUF variant did not bind to the wild-type RNA (Fig. 7A, middle panel). These results validate that the TRM combinations identified in the three-hybrid assay reflect changes in binding specificity that can be detected biochemically in gel shift experiments.

We purified the design 1 variant (with changes to two TRM combinations but only one change in RNA base recognition) and design 2 variant (with both two TRM combinations changed and two RNA base changes in the target) in order to test their in vitro binding in the electrophoretic mobility shift assay. Design 2 showed sequence specificity, binding to its target RNA sequence

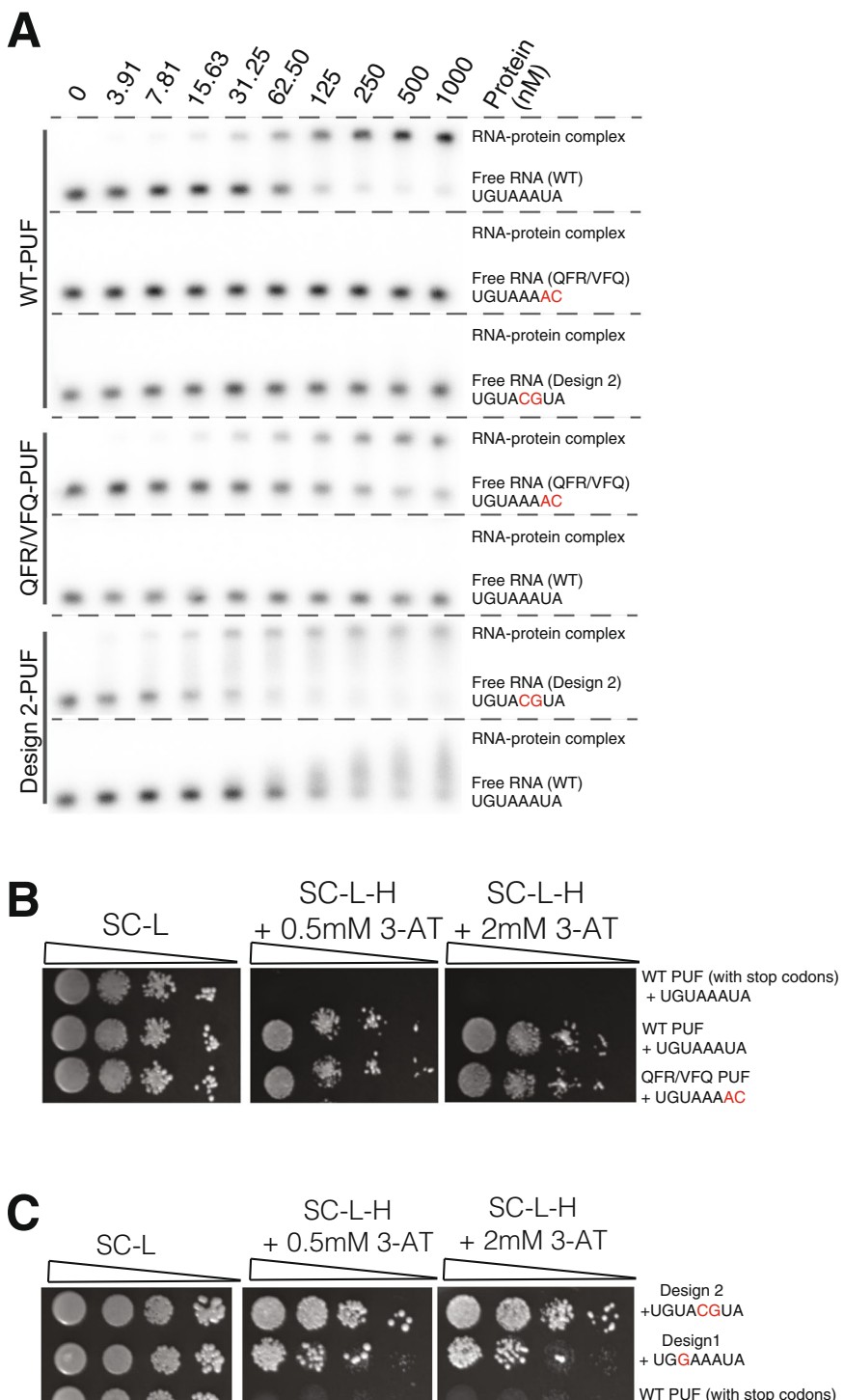

**Fig. 7 Electrophoretic mobility shift assays and three-hybrid assays for wild-type and variant PUF domains. A** Electrophoretic mobility shift assays show the in vitro binding for the wild-type PUF domain, the QFR/VFQ PUF variant, and the design 2 variant to the wild-type RNA sequence or an RNA containing mutated bases. Results are representative of two biological replicates. **B** Yeast three-hybrid assays show the binding of the wild-type PUF domain and QFR/VFQ variant. The negative control is a wild-type PUF domain that has stop codons in the TRM locations (repeat 1) paired with wild-type RNA. SC-L, synthetic complete media minus leucine; SC-L-H, synthetic complete media minus leucine and histidine. **C** Spot dilution plate assay indicates the binding of the design 1 and design 2 PUF domains to their target RNAs. The negative control is as in (**B**). The starting OD600 that was spotted was 0.05, with three sequential 10-fold serial dilutions shown.

but not to the wild-type RNA sequence. Specific binding occurred with an approximate Kd for this pair of ~45 nM (95% CI 33–62 nM). Heterogeneity of the shifted bands may indicate dissociation during electrophoresis or non-specific interactions. Design 1 did not bind to either its target or the wild-type RNA sequence in the gel shift assay. Both the design 1 and design 2 PUF variant proteins showed binding in a yeast three-hybrid plate assay, although the results of the dilutions indicated that the design 1 variant had about a 10-fold weaker signal than the design 2 variant (Fig. 7C). For design 1, the yeast three-hybrid assay may be more sensitive to identify a protein-RNA interaction than an in vitro binding assay.

We attempted to purify PUF variants with three or more mutated repeats (designs 8, 15, and 16), but these proteins, containing nine or more altered residues, were insoluble. Others have also observed that variant PUF proteins have been difficult to obtain in soluble form from _E. coli_[30,47,48]. For example, Cheong and Hall[30] were unable to produce the soluble protein of human Pumilio-1 when they mutated the residue in repeat 7 or 3 that forms a stacking interaction with the base. Though future studies would be needed to examine the correlation between the three-hybrid selections and affinity, the values we determined in the biochemical assays generally corroborated behavior in the yeast selections.

## Discussion

The PUF domain's modular architecture of eight repeats and its affinity and specificity for binding to an 8-mer RNA sequence make it attractive for engineering a protein to bind to an arbitrary RNA sequence. We elucidated the RNA-binding preferences of nearly all possible TRM combinations for each of the repeats of the human Pumilio-1 PUF domain. By calculating scores both for binding interaction and for specificity, we obtained base-specific recognition patterns for each repeat as a resource for the rational design of PUF domains (Fig. 4). For some repeat locations, we did not identify an optimal TRM variant for RNA binding. One potential reason is that not all candidate TRM variants were tested in each repeat due to cloning bias. Another is that some promising TRM variants did not display sufficient specificity across all repeats. Expanding the yeast-three-hybrid libraries to explore the inter-domain interactions between repeats and the contribution of non-TRM residues to binding may facilitate the identification of novel repeat combinations that will fill in the missing gaps. In principle, reiteration of pairs of PUF repeats might minimize context effects and so simplify the recognition code across repeats. RNA-based targeting via Cas13 systems provides a powerful alternative when applicable, and PPR strategies, which rely on reiterated protein repeats for recognition, also have great promise.

Many base-specific TRM combinations identified for one repeat are not generic across all repeats, suggesting that distinct codes should be considered depending on the repeat, and the protein being engineered. For example, previously described C-specific codes (e.g., SYR and CYR)[34] showed binding when present in a few, but not all, repeat locations. We identified novel TRM combinations that worked well for RNA base recognition in each repeat of the PUF domain. However, context effects apparent from our selections complicate the application of PUF proteins in rational design. Variations among the specificity code might be minimized by using a scaffold derived from a single repeat, or pairs of repeats, in series, rather than a naturally occurring PUF protein. Such derivatives could be assessed using assays described here.

We found that N-terminal locations in a PUF domain were more robust for tolerating combinatorial mutations than

C-terminal locations, as observed previously[28]. In addition, crosstalk appeared to occur between N- and C-terminal repeats. For example, while double variants with substitutions in only one terminus worked well in recognition of a new RNA target, the addition of one or two substitutions at the other terminus of the domain often resulted in failed designs. This phenomenon might be related to PUF domain structure, with basic concave and acidic convex surfaces critical for RNA binding and structural stability[27]. Mutations in both N- and C-terminal repeats might lead to the partial unfolding of the domain due to changes in surface acidity. Another possibility is that two PUF domains may bind to a single RNA sequence in an antiparallel fashion. Gupta et al.[49] reported that two PUF domains can co-occupy a single intact NRE RNA with cooperative binding, and that this phenomenon can be found in other non-canonical PUF proteins (e.g., yeast Puf2 protein)[50]. Thus, beyond a focus on designing specificity for each individual repeat location in an engineered PUF domain, crosstalk between repeat locations should be considered to maximize affinity.

Campbell et al.[28] scored the prevalence of TRM combinations at each PUF repeat in 94 Pumilio-1 homologues, inferring the abundance of natural TRM combinations from the sequence alignments. Their aligned data are broadly consistent with our high-throughput screening results. It is striking that the C-terminal repeats of the PUF domain can be rationally designed to bind other RNA sequences, yet are highly conserved in their specificity during evolution. As has been noted[28], the observation that a wide range of PUF proteins maintain similar C-terminal TRMs and a UGU sequence at the 5' end of the binding site implies that this region executes biological roles beyond RNA binding that constrain the protein's evolutionary divergence.

Stacking residues in each repeat play an important role in specificity. Stacking interactions are pervasive in PUF domain-RNA complexes, resulting from columns of stacking bases and amino acid side chains along the entire length of the PUF protein. Koh et al.[51] found that stacking amino acids in the _C. elegans_ FBF-2 PUF domain contribute to the protein's specificity for its RNA sequence. The structural analysis supports the idea that different stacking arrangements can lead to different specificities for RNA recognition[51]. Campbell et al.[28] provide additional examples of the role of stacking interactions. By comparing base-specific recognition patterns across repeats, we found that stacking interactions at different positions influence specificity. For example, SWD specifically recognized guanine in repeats 5, 6, and 7, while SGD was specific for guanine in repeats 1 and 3. These results support the idea that the stacking residue in each repeat may function in combination with neighboring amino acids to specify an RNA base. Koh et al.[51] tested whether the identity of the stacking residue contributes to specificity for the neighboring 3' base and found that the effect is limited. Our data suggest a similar conclusion. For example, with mutation of SNE to NPG in repeat 7, the substitution from asparagine to proline in the stacking residue did not alter the preference for U at the neighboring 3' base (base 3) (Fig. 6). The same phenomenon can be seen in other designs as well, which indicate a limited nearest-neighbor effect resulting from the stacking residue substitution.

Finally, our findings indicate a balance between individual binding specificity and total binding affinity. In the evolution of PUF proteins, TRM combinations have been selected to increase or decrease individual repeat specificity while maintaining the total binding affinity needed for biological function[51–53]. Structural studies have found that several PUF proteins exhibit broader specificity through the ejection of certain "undesirable" nucleotides[54]. This mechanism can provide a basis for PUF recognition of degenerative binding sites and can greatly increase the number of RNA targets in vivo. For example, some PUF

proteins (e.g., yeast Puf4) use their eight repeats to bind to RNA sequences with nine or ten bases by allowing one base to be turned away from the RNA-binding surface[55]. Occasionally, base flipping can occur to accommodate simultaneous occupancy of the binding pockets. Wang et al.[36] suggest that PUF proteins likely exist with greater flexibility to allow base flipping to accommodate different modes of binding to achieve overall binding affinity. With these possibilities in mind, a multi-stage process should be considered for engineering PUF domains to bind to a target RNA. Initially, PUF designs having TRM combinations in each repeat specific for the target RNA can be used in screens to recover RNA targets that bind. Then, this information can be incorporated to determine additional design features for improved binding. The addition of machine learning to the screening results may enable features critical for PUF-RNA binding to be readily exploited.

## Methods

**Generation of "all-in-one" construct and TRM libraries**. A centromeric vector designated pAIO3H was created by cloning the NotI-RNA module-NotI fragment from p3HR2 (ref.[46,56]) into the NotI site of pACT2 (ref.[37,46]). Unique restriction sites for cloning the protein-coding sequence and the RNA element were added to the construct. With the Pumilio-1 PUF domain cloned into NcoI and SacI sites of pAIO3H, either the wild-type p38α-NREa or each of its possible single-base substitutions variants were cloned into the XmaI and SphI sites of the RNA expression module, generating 25 distinct pAIO3H plasmids. For the randomized libraries, we used 32 primers (four primers for each PUF repeat), with each primer containing a random NNK (K = G or T) for each of the three TRM residues. In addition, we introduced to each primer a two-base code in the form of synonymous changes in codons 11 and either 14 or 15, which are adjacent to the TRM residues, to specify the identity of the RNA base in the cognate binding site. Each of the PCR products that included the randomized TRMs and the RNA base-specifying synonymous changes was cloned into one of the 25 pAIO3H plasmids with the matching RNA mutation, thereby allowing the identification of the protein-RNA partners by sequencing the region encoding the TRM and adjacent residue. For the targeted library, we ordered an oligo pool that contains 2000 fragments (Twist Bioscience) and incorporated them into the construct through Gibson assembly[57].

**Yeast three-hybrid screen**. Library plasmids were transformed according to a previous protocol[58]. The yeast strain constitutively expresses the fusion protein LexA–MS2 coat and a HIS3 reporter gene under the control of multiple LexA operators. The genotype of YBZ-1 is MATa, ura3-52, leu2-3, 112, his3-200, trp1-1, ade2, LYS2:: (LexAop)-HIS3, ura3:: (lexA-op)-lacZ, LexA-MS2 coat (N55K). We collected transformants from plates containing SC-Leu media[37]. For the three-hybrid selection, we plated the transformants onto plates containing SC-Leu media and plates with SC-Leu-His + 0.5 mM 3-AT media. Colonies were grown for four days. We collected cells and extracted their plasmids (Zymoprep Yeast Plasmid Miniprep II kit; Cat 11-315).

**Sequencing library preparation and analysis**. The region including each PUF repeat was amplified through sequential reactions. Internal PCR was carried out through primers with sequences that anneal to each repeat (primers 623–654) and external PCR was carried out to add an Illumina sequencing adapter (primers 419–422), which is provided in Supplementary Data 3. Phusion polymerase was used for these reactions, and each reaction was performed on a BioRad Mini-Opticon and monitored to avoid over-amplification. The PCR products were sequenced using the Nextseq 550 platform. Downstream analyses were performed in R.

**Illumina sequencing reads processing**. Base calls were converted to fastq format and demultiplexed using Illumina's bcl2fastq/2.16.0.10 tolerating one mismatched base in barcodes (edit distance (ED) < 2). Read1 and read2 were merged using SeqPrep/2016. Low quality reads were filtered based on the exact match of the first 10 bp common sequence in the plasmid. Reads were first filtered by their internal common sequence, with those that matched this sequence retained and those that did not discarded. We then divided the matched reads into eight groups and assigned a repeat identifier to each matched read based on the unique common sequence in each repeat. The filtered reads were then trimmed to a 16 bp-sequence. The first base of this sequence indicates RNA identity (A/G/U/C) and the following 15 bp indicate the amino acid sequence from position 12 through position 16. Based on the amino acids in position 12, 13, and 16, we assigned a PUF variant identifier to that read. Sequencing read counts corresponding to a given PUF variant were equal to the sum of read counts from all trimmed reads matching that variant.

**Generation of interaction scores and specificity scores for PUF variants**. We determined the frequency of each variant in both input pool and post-selection pool by comparing its sequencing reads from both pools. The log2 change in the frequency from input to selection pool serves as a measure of binding activity for each PUF variant, designated as a PUF domain–RNA interaction score in this assay. For those PUF variants only shown in the input pool, their interaction score is calculated by adding a count of 1 (pseudocount) in the post-selection pool. A specificity score is assigned to each PUF variant calculating the difference between its highest and second-highest interaction score. For Fig. 5, we obtained normalized scores for each base as follows. If the interaction score of a PUF variant to A, C, G, and U is X1, X2, X3, X4, assume X4 is the maximal value among them. Then, the normalized score against A, C, G, U was calculated as (X1/X4), (X2/X4), (X3/X4), (X4/X4).

**Generation of sequence logos to summarize base-specific recognition patterns**. For PUF variants with interaction scores >0, we carried out unsupervised hierarchical clustering analysis based on the scores against all four RNA bases. PUF variants with similar RNA specificity were clustered together. We focused on regions of variants with specificity >4 and generated sequence logos[59] to summarize the recognition patterns. For regions with fewer than 10 variants, we used one sequence logo to summarize all information. For regions with more than 10 variants, we manually separated these variants into several subgroups based on the properties of the amino acids (e.g., positively or negatively charged or neutral) in the three positions and generated separate sequence logos to summarize the patterns of these subgroups.

**Generation of RNA oligo array for identifying combinatorial effects**. We designed the RNA oligo array with 1900 RNA elements; 900 of them comprised 20 copies of wild-type sequence and 55 copies of each of the 16 target sequences. For the remaining 1000 RNA sequences, we programmed the percentage of A, C, G, and U in each location of the 8-base RNA sequences based on a combination of the 16 targets, such that for each location, the programmed base had a relatively high probability to match the target bases and a low probability to match the off-target bases. We used the code: base_1 = sample(x = c("A", "T", "G", "C"), size = 1, prob = c(0.10, 0.70, 0.10, 0.10)); base_2 = sample(x = c("A", "T", "G", "C"), size = 1, prob = c(0.10, 0.49, 0.31, 0.10)); base_3 = sample(x = c("A", "T", "G", "C"), size = 1, prob = c(0.10, 0.35, 0.45, 0.10)); base_4 = sample(x = c("A", "T", "G", "C"), size = 1, prob = c(0.70, 0.10, 0.10, 0.10)); base_5 = sample(x = c("A", "T", "G", "C"), size = 1, prob = c(0.49, 0.10, 0.10, 0.31)); base_6 = sample(x = c("A", "T", "G", "C"), size = 1, prob = c(0.35, 0.10, 0.45, 0.10)); base_7 = sample(x = c("A", "T", "G", "C"), size = 1, prob = c(0.35, 0.45, 0.10, 0.10)); base_8 = sample(x = c("A", "T", "G", "C"), size = 1, prob = c(0.54, 0.10, 0.10, 0.26)).

**Spot dilution plate assay**. Cells were grown overnight in YPD medium at 30 °C. The cultures were diluted to an OD600 of 0.05, and 3 additional 10-fold serial dilutions were made. The cells were spotted onto plates with one of three different media: synthetic complete without leucine (SC-L); synthetic complete without leucine and histidine and with 0.5 mM 3-AT (SC-L-H + 0.5 mM 3-AT); and synthetic complete without leucine and histidine and with 2 mM 3-AT (SC-L-H + 2 mM 3-AT).

**Protein expression and purification**. The wild-type and PUF variants were subcloned into the pGEX-6P-1 plasmid to generate an N-terminal GST tag using the Gibson assembly method. Briefly, the vector backbone was digested with EcoRI-HF (NEB, R3101L) and BamHI-HF (NEB, R3136L) enzymes, and the insert with complementary overhang sequences was amplified using Phusion polymerase and cleaned up using the Zymo DNA Clean & Concentrator™ Kits (Genesee Scientific, 11-303). Gibson assembly was performed according to the manufacturer's instructions (NEB, E2611L) and used directly to transform One Shot™ TOP10 Chemically Competent E. coli cells (Thermo Fisher, C404003) by heat shock at 42 °C for 45 s. The resulting GST fusion proteins were expressed in BL-21 E. coli cells, which were grown at 37 °C to O.D. 0.6, chilled on ice, and incubated at 16 or 18 °C overnight after the addition of IPTG to 400 μM final concentration. Cells were harvested by centrifugation, washed with PBS, and stored at -80 °C overnight. Protein purification was performed using B-PER™ Complete Bacterial Protein Extraction Reagent (Thermo Fisher, 89821) following the manufacturer's instructions and the addition of 1 mM EDTA, 1 mM DTT, and 1x Complete™ Protease Inhibitor (Sigma, 11873580001). Cell extracts were purified using either Pierce™ Glutathione Chromatography Cartridges (Thermo Fisher, 16109) (wild-type, QFR/ VFQ variant and design 3) or Pierce™ GST Spin Purification Kit (Thermo Fisher, 16106) (designs 1, 2, 8, 5, 15, and 16). Glutathione agarose was washed with Bind/ Wash buffer (50 mM Tris pH 8.0,150 mM NaCl, 1 mM EDTA, 1 mM DTT) and cell lysates mixed with Bind/Wash buffer at 1:1 ratio were loaded onto Glutathione columns by syringe or incubated while rotating at 4 °C for 1 h. Bound protein was washed three times using Bind/Wash buffer and on-column or on-resin cleavage was performed by adding Elution buffer (50 mM Tris pH 8.0,150 mM NaCl, 1 mM EDTA, 1 mM DTT, 1% Triton-X) containing Turbo3C (HRV3C) Protease

(Biovision, 9206-1). Purified proteins were analyzed using SDS-PAGE and protein concentrations were determined using the Bradford assay.

**Electrophoretic mobility shift assays**. 20-mer Oligoribonucleotides were ordered from IDT. The NRE was flanked by six upstream and six downstream bases (e.g., wild-type RNA is rGrGrUrArArGr*UrGrUrArArArUrA*rGrUrCrUrGrCrArU). The 20-mer oligoribonucleotides were labeled with [$\gamma$-32P]-ATP and T4 polynucleotide kinase (NEB, M0201S) using standard conditions. The labeled RNAs were denatured in 1X EMSA buffer (10 mM HEPES pH 7.3, 50 mM NaCl, 2 mM DTT, 0.05% Tween 20, 0.1 mg/ml BSA, and 0.1 mg/ml yeast tRNA) without BSA and tRNA by heating the sample for 30 s at 80 °C and cooling it down on ice for 2 min[60]. About 10 nM labeled, denatured RNA was incubated with 0–1 μM protein in 14-μl reactions in 1× EMSA buffer. Reactions were incubated on ice for 1 h. After the addition of 5 μl 4× loading buffer (1× EMSA buffer with 25% glycerol, 0.1% Xylene Cyanol FF, and Bromophenol Blue), samples were separated on a 1.2% agarose gel for 1 h at 100 V at room temperature using 1× TBE as running buffer. The agarose gel was dried, exposed to a phosphorimager screen overnight and radioactive bands were detected using the Typhoon FLA 9500 biomolecular imager (GE Healthcare)[61]. The Kd value is the concentration of PUF at which 50% of the RNA is in a complex with the protein. It was determined by first using ImageJ software to quantify the signal in each RNA band. Background signals from blank regions of the gel were subtracted from the signal intensities obtained from the bands. The fraction of RNA bound was determined from the background-subtracted signal intensities using the expression: bound/(bound + unbound). The fraction of RNA bound in each reaction was plotted versus the concentration of PUF protein. We then used Prism Software to perform non-linear regression and obtain a value for Kd and its 95% CI.

**Reporting summary**. Further information on research design is available in the Nature Research Reporting Summary linked to this article.

## Data availability

The data supporting the findings of this study are available from the corresponding authors upon reasonable request. The high-throughput sequencing reads, along with datasets showing calculated interaction scores for both random screening and targeted screening generated in this study have been deposited in the Gene Expression Omnibus (GEO) under accession code GSE152452. Source data are provided with this paper. In addition, the crystal structure of the Pumilio-homology domain and RNA interaction is available online in database PDB ID 1M8Y. Source data are provided with this paper.

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

## Acknowledgements

We thank Deena Oren (The Rockefeller University, Structural Biology Resource Center) for her help with the protein purification design and for providing the necessary equipment. We thank Yoav Arava (Israel Institute of Technology) for discussions about this project. This work was supported by grant P41 GM103533 from the National Institutes of Health (to S.F.) and CEGS grant RM1 HG011014 from the National Institute of Health (to J.C.).

## Author contributions

W.Z., D.M., M.W., and S.F. designed the research. D.M built the initial TRM libraries, and W.Z built the other libraries and performed the screen experiments. W.Z performed the computational analysis with the help from J.C. G.B. purified the proteins for the in vitro assays under the supervision of J.C. C.M. performed the EMSA assays under the supervision of T.T. W.Z, D.M., M.W., J.C., and S.F. wrote the paper.

## Competing interests

The authors declare no competing interests.

## Additional information

**Peer review information** *Nature Communications* thanks Ian Small and other, anonymous, reviewers for their contirbutions to the peer review of this work. Peer review reports are available.

