## [Peer Review File · Nature Communications]

Title: Expanding the binding specificity for RNA recognition by a PUF domainREVIEWER COMMENTS

Reviewer #1 (Remarks to the Author):

PUF proteins have highly desirable properties that enable their design as RNA-binding proteins that target RNAs of interest. In this manuscript Zhou and colleagues explore the code for RNA recognition by PUFs in greater detail than any previous study, providing an important resource for future efforts in this area. The work is compelling, solid and should be of interest to the scientific community.

Key issues to be addressed before publication fall into two areas:

1. Details of the library selections

- (a) The wild-type PUM1 target is described as 5'UGUAAAUA3', however, the studies of the Hall group (PMID: 12202039) supported a 5'UGUA(U/C)AUA3' motif by structural and biochemical criteria. I can see that an A at position 5 is supported by the data in Figure S2 but I wonder if some discussion of the choice of this starting sequence might be valuable. Could the choice of neighbouring nucleotides influence the nature of the selected protein motifs?
- (b) Although most stop codons were depleted in the selections, some appeared to be enriched. Does this indicate that some truncated PUF proteins might have residual RNA-binding activity or does this simply represent the level of noise in the selections?
- (c) I could not find Figure S4 in the materials provided.
- (d) Was there a bias in the selection of targeted motifs for the second round of selections? I note the loss of the A-binding motif (IFV) from repeat 1 as one example. Also VFR was the only motif with an R at position 16 tested in the second round for repeat 2, potentially explaining the lack of a C-binding motif for this repeat in the final data. There are a number of these instances that should be discussed, so readers can understand what might have been represented/missing in these data.
- (e) The methods section describing the deep sequencing analyses should be expanded. How were reads trimmed and aligned; which parameters were used? What was the typical coverage achieved?

2. Coverage of the literature and referencing

There is an established literature on the use of RNA-binding proteins to target RNAs of interest, however, much of this has been covered too sparsely. Nature Communications allows up to 70 references, while the current manuscript uses only 32. To fairly represent the literature and also provide an adequate background to readers who might not be as familiar with this field, the authors should expand the introduction and discussion to include, but not limited to, the following important contributions.

- (a) Regarding RRM (lines 56-59) important work from Allain and Varani should be mentioned (e.g. PMIDs 28935965 and 27428511).
- (b) For PPR (lines 60-62), in addition to the cited references, work from the following established the code for RNA recognition and its portability: PMIDs 23472078, 25517350 and 27088764.
- (c) Regarding Cas13 (lines 62-64), the key papers to cite are PMIDs 27256883, 29551514 and 28065598.
- (d) For PUF protein recognition of RNA (lines 67-76), the following important early studies should be cited: PMIDs 9888799, 11336677.

- (e) The citation for reference 13 on line 86 should be for Cheong et al. not “Cheone et al.”.
- (f) Regarding crosstalk between N-terminal and C-terminal repeats (lines 369-381), could this relate to the cooperative binding of multiple PUF proteins observed by Gupta et al. (PMID: 19372537)?
- (g) In the discussion, regarding the influence of individual repeats versus overall binding affinity the work of Adamala et al. using multiple identical PUF repeats (PMID: 27118836) and Wallis et al. using increased affinity to rescue impaired stability of a PUF (PMID 29808990) should be included.
- (h) In the discussion, for the results of this study and PUF design in general could the authors please consider how the results of Gupta et al. (PMID 18328718) and Wang et al. (PMID 19901328), where nucleotides in bound RNAs were observed to flip out of the protein binding site, might need to be considered.
- (i) The reference for Gibson assembly is an online manual from NEB (reference 30). It would be more appropriate to cite the original scientific paper: PMID 19363495.

Reviewer #2 (Remarks to the Author):

The submitted manuscript by Zhou et al. describes a yeast three-hybrid approach combined with next generation sequencing to expand the RNA-binding specificity of a PUF domain, which comprises a modular architecture of eight repeats binding to an RNA 8-mer sequence. As such, it provides a bona fide target for the design of an RNA-binding domain with programmable specificity and applicability in research and therapies. The authors here combine randomizing both the PUF domains TRM in each of the eight repeats and any of the four bases for the cognate RNA sequence position. Their data reveal not only an extended set of TRM combinations for either of the bases, but also show a repeat-specific sequence pattern. From that, Zhou et al. derived an extended RNA-binding code considering each repeat and base. This code has then been challenged against a set of differentially constructed target RNAs, for which cognate domain versions have been tested, incl. multiple changes within the RNA sequence with respect to the wildtype.

Altogether, I highly appreciate the systematic, initially unbiased approach underlying the non-trivial scientific challenge. The herein used Y3H system may not be able to fully cope with the complexity, but the analysis of data seems straightforward and allows for a full description of sequence preferences on both the protein and RNA sides. The findings are in principle trustworthy, convincing, the data presented in a comprehensible way, even for a non-specialist and despite the complex genetic setup. As of its current stage, the manuscript however appears to be a bit more of a resource and will require some more validation of the suggested RNA-protein code. I would thus suggest the manuscript for publication, but for this, some additional points will have to be addressed by the authors.

For a broader exploitation of findings, I still see 2-3 major necessities of additional data:

- While I do see the gain in new information from the presented data, and possible benefits from them, I am skeptical about the sole relying on the two scores for interaction and specificity. The authors should

revalidate some of the major findings with experiments that allow for an alternative quantification of interactions between PUF and the RNA 8-mer. They might e.g. pick a few of their designed PUF domains, produce them recombinantly, and measure differential affinities with the cognate RNAs via techniques like SPR, ITC or similar ones, including the WT references (both protein and RNA), and control mutants. Those measurements will support the concept of real specificity outside the Y3H context and help to estimate the range of affinities in the context of the claimed expanded specificity.

- In light of the above-mentioned and what I did not get from the manuscript: Do the authors also find a new combination of PUF-RNA that suggests a higher affinity than the WT-WT pair? This should be used in an in vitro assay as well.

- Seeing the strong benefit from repeat-resolved specificity for new PUF design, what would the authors suggest as a potential design pipeline, e.g. assuming a random 8-mer to be targeted with a perfect PUF module?! How would this outdo previous rational designs that were simply based on TRM specificity per base. Could the authors use the in vitro binding assay with a suitable example RNA?!

I also have some minor points to be addressed:

- There is no Supplementary Figure S4 as part of the Supplement although it is referred to in the text on page 7.

- I was not able to identify Supplemental Files 1 and 2. Not sure, but they were not accessible for me.

- I might have missed this, but could authors please explain the ratio for the choice of the four TRMs per base and repeat position in Figure 5?!

- Considering the seemingly important role of the sequence context: wouldn't one have to expect a yet bigger influence of flanking RNA sequences (beyond the 8-mer core), taking into account they might (accidentally) provoke a register shift along repeats, modulate local RNA backbone geometry, base stacking that extends into the flanking nucleotides etc. and thus influence the precise fit of a designed PUF domain, especially in a potential application? The authors have made all their conclusions from sequences in an identical context. While this is of course necessary within the experimental logics, the effects of differential flanking sequences, e.g. comparing the 5' vs. 3' halves' robustness to changes, might not be neglectable. The authors should comment on that!

Reviewer #3 (Remarks to the Author):

This manuscript describes an analysis of the RNA binding specificity of the human Pumilio-1 protein, a PUF protein that has already been the focus of previous studies. This new study is considerably more systematic and detailed than any of the preceding ones and provides a lot of data on the base preferences of the motifs at different positions in the protein. Although this is an impressively thorough and careful piece of work that will be of great help to researchers on PUF proteins, I have some doubts about the broader general interest of the study as the results tend to indicate that this may not be the

best way forward if the goal is to develop 'the ability to design a protein that can bind specifically to a target RNA and regulate its fate'.

1. The focus of the manuscript is on the potential use of this data to rationally design PUF proteins to bind novel target sequences, and some successful demonstrations are included. However, the conclusion reached by readers of this work is likely to be that PUF proteins (or at least Pumilio-1) are relatively poor candidates for such rational design. This is due to the prevalence of position-specific effects (the base recognition 'code' differs at different positions, Figs 3-5) and apparent long range interactions between distant motifs (changes at one end of the protein affect binding at the other end, Fig 6 and Discussion). The result is that only relatively minor differences to the target sequence can be tolerated whilst still achieving strong and predictable binding. The ease of design and the predictability of binding is certainly lower than for Cas13 and related RNA-guided proteins, and even appears to be worse than for PPR proteins, which can be relatively easily designed to bind novel targets (e.g. Miranda et al *Nucleic Acids Research*, 46: 2613–2623; Colas des Francs-Small et al. *Communications Biology*, 1: 166; Yan et al. *Nucleic Acids Research*, 47: 3728–3738; McDermott et al. *Plant Cell*, 31:1723–1733). Cas13 and PPR proteins are only mentioned in passing (and summarily dismissed) in the introduction; a discussion of their relative merits would be appropriate in the discussion.

2. I suspect that some (many?) of the complexities encountered in this work are due to the decision to start with a natural PUF protein (human Pumilio-1) rather than a synthetic protein based on quasi-identical consensus motifs. The latter approach has proved to be a very powerful means to simplify and improve rational design of both PUF and PPR proteins. It seems likely that this would reduce position-specific effects (and, less likely, the long range interactions). I would like to see a justification for the approach that was chosen.

3. A potential advantage to the use of a natural protein is the ability to make use of millions of years of evolution in the exploration of the sequence space. This advantage is not used here. A discussion of sequence variability in Pumilio-1 homologues and their target sites might nicely complement the experimental results.

4. Concerning the data itself, everything in manuscript comes from a rather indirect assay, namely a high-throughput three-hybrid screen in yeast cells. Although the data look good and I see no particular issues with these experiments, I would have liked to see some confirmation by more direct biochemical means (e.g. in vitro measurements of binding affinity and specificity by one of ITC, MST, FP, SPR, REMSA...)

5. The explanations proposed for position-specific effects appear to focus on possible interactions within the PUF motifs (or between neighboring motifs). I wonder if the authors have looked at possible nearest-neighbor effects in the RNA? As stacking interactions are clearly significant for binding and specificity, the nature of the neighboring 3' nucleotide (especially whether it is a purine or pyrimidine) might need to be taken into account. Currently, the fact that the amino acid sidechain at position 13 contacts two bases doesn't appear to be considered. Even if it can be easily eliminated as a factor it

would be worth discussing.

6. When the natural site is modified, can we be sure that the protein variants will bind to exactly the same position? All the analyses are assuming no shift in binding, but it's not clear how safe this assumption is.

Reviewer #1 (Remarks to the Author):

PUF proteins have highly desirable properties that enable their design as RNA-binding proteins that target RNAs of interest. In this manuscript Zhou and colleagues explore the code for RNA recognition by PUFs in greater detail than any previous study, providing an important resource for future efforts in this area. The work is compelling, solid and should be of interest to the scientific community.

Key issues to be addressed before publication fall into two areas:

1. Details of the library selections

(a) The wild-type PUM1 target is described as 5'UGUAAAUA3', however, the studies of the Hall group (PMID: 12202039) supported a 5'UGUA(U/C)AUA3' motif by structural and biochemical criteria. I can see that an A at position 5 is supported by the data in Figure S2 but I wonder if some discussion of the choice of this starting sequence might be valuable. Could the choice of neighbouring nucleotides influence the nature of the selected protein motifs?

We chose the sequence UGUAAAUA as the starting sequence based on an available structure (Lu and Hall 2011), the use of this sequence in other studies (e.g. Galgano et al. 2008) and our three-hybrid results.

We added to the text (p. 6):

“We chose the NRE element UGUAAAUA as the starting recognition sequence rather than a sequence with U or C at position 5 (ref. 26), as UGUAAAUA is the most common Pumilio-1 binding motif based on a comparative analysis of mRNA targets for human PUF family proteins (ref. 39), a structure exists for human Pumilio-1 bound to this sequence (ref. 30), and A was one of three preferred bases at that position when Pumilio-1 binding was tested in the yeast three-hybrid assay (Figure S2).”

(b) Although most stop codons were depleted in the selections, some appeared to be enriched. Does this indicate that some truncated PUF proteins might have residual RNA-binding activity or does this simply represent the level of noise in the selections?

This type of noise is routinely seen in other deep mutational scanning experiments (e.g. Matreyek et al. 2018).

We added to the text (p. 8):

“Some nonsense variants had scores that indicated they were enriched, which may result from experimental noise, as routinely seen in other deep mutational scanning experiments (ref. 42);

the nonsense variants with these enrichment scores had significantly lower input reads than other nonsense variants. Use of these scores allowed us to calculate a false positive rate for loss-of-function missense mutations. We found that 1.4% (45/3193) of nonsense variants had an enrichment score >0 , providing an estimate of the fraction of the loss-of-function missense variants that were also false positives.”

Here are the read counts:

(c) I could not find Figure S4 in the materials provided.

Sorry for the missing figure. Supplementary figure 4 has been added to the manuscript.

(d) Was there a bias in the selection of targeted motifs for the second round of selections? I note the loss of the A-binding motif (IFV) from repeat 1 as one example. Also VFR was the only motif with an R at position 16 tested in the second round for repeat 2, potentially explaining the lack of a C-binding motif for this repeat in the final data. There are a number of these instances that should be discussed, so readers can understand what might have been represented/missing in these data.

We agree that for some repeat locations, we did not identify an optimal TRM variant that can be used for RNA binding.

From the initial randomized yeast three-hybrid screen, we selected certain promising candidates by setting a threshold of interaction score and specificity score. The number of candidates we selected from each repeat ranged between 181 and 299 (Supplemental file 1). We synthesized an oligonucleotide pool that includes all the candidates. To survey the binding activity score against all four RNA bases and across all eight repeats, we manually cloned this oligo pool into each repeat location in order to survey the same TRM variants across all eight repeat locations. Therefore, VFR was not the only motif with an R at position 16 tested in the second round for repeat 2.

However, there are cloning biases during this process. For example, as shown below (part of Figure 5), CFP and VFQ are promising A-binding TRM combinations, but VFQ was not cloned

into repeat 3 and repeat 5; and CFP was not cloned into repeat 3. We would need to redo the experiment to increase the transformation efficiency or clone promising TRM combinations individually to test their binding activity.

2) Many promising TRM combinations were specific in some repeats but non-specific in others. This factor also caused us to not identify a code in some repeats. For example, VFR was not the only code with R tested for repeat 2, as QWR was tested as well, but it was not as specific for repeat 2 (shown below; Figure 5).

We have added to the Discussion (p. 19):

“For some repeat locations, we did not identify an optimal TRM variant for RNA binding. One potential reason is that not all candidate TRM variants were tested in each repeat due to cloning bias. Another is that some promising TRM variants did not display sufficient specificity across all repeats. Expanding the yeast-three hybrid libraries to explore the inter-domain interactions between repeats and the contribution of non-TRM residues to binding may facilitate the identification of novel repeat combinations that will fill in the missing gaps. In principle, reiteration of pairs of PUF repeats might minimize context effects and so simplify the recognition code across repeats.”

(e) The methods section describing the deep sequencing analyses should be expanded. How were reads trimmed and aligned; which parameters were used? What was the typical coverage achieved?

We added another Methods section describing the details of sequencing reads processing.

2. Coverage of the literature and referencing

There is an established literature on the use of RNA-binding proteins to target RNAs of interest, however, much of this has been covered too sparsely. Nature Communications allows up to 70 references, while the current manuscript uses only 32. To fairly represent the literature and also provide an adequate background to readers who might not be as familiar with this field, the authors should expand the introduction and discussion to include, but not limited to, the following important contributions.

We added the mentioned references to the manuscript. We now provide in the Introduction a more detailed description of other RNA-binding proteins and their advantages and disadvantages.

(a) Regarding RRM (lines 56-59) important work from Allain and Varani should be mentioned (e.g. PMIDs 28935965 and 27428511).

We added these references to the manuscript.

(b) For PPR (lines 60-62), in addition to the cited references, work from the following established the code for RNA recognition and its portability: PMIDs 23472078, 25517350 and 27088764.

We added these references to the manuscript.

(c) Regarding Cas13 (lines 62-64), the key papers to cite are PMIDs 27256883, 29551514 and 28065598.

We added these and other Cas13 references to the manuscript.

(d) For PUF protein recognition of RNA (lines 67-76), the following important early studies should be cited: PMIDs 9888799, 11336677.

We added these references to the manuscript.

(e) The citation for reference 13 on line 86 should be for Cheong et al. not “Cheone et al.”.

We corrected this typo.

(f) Regarding crosstalk between N-terminal and C-terminal repeats (lines 369-381), could this relate to the cooperative binding of multiple PUF proteins observed by Gupta et al. (PMID: 19372537)?

We added this reference and now provide more discussion in the manuscript regarding the crosstalk between N-terminal and C-terminal repeats. It reads (p. 20):

“This phenomenon might be related to PUF domain structure, with basic concave and acidic convex surfaces critical for RNA binding and structural stability (ref. 27). Mutations in both N- and C-terminal repeats might lead to partial unfolding of the domain due to changes in surface acidity. Another possibility is that two PUF domains may bind to a single RNA sequence in an antiparallel fashion. Gupta *et al.* (ref. 49) reported that two PUF domains can co-occupy a single intact NRE RNA with cooperative binding, and that this phenomenon can be found in other non-canonical PUF proteins (*e.g.* yeast Puf2 protein, ref. 50). Thus, beyond a focus on designing specificity for each individual repeat location in an engineered PUF domain, crosstalk between repeat locations should be considered to maximize affinity.”

(g) In the discussion, regarding the influence of individual repeats versus overall binding affinity the work of Adamala et al. using multiple identical PUF repeats (PMID: 27118836) and Wallis et al. using increased affinity to rescue impaired stability of a PUF (PMID 29808990) should be included.

We added this reference and now provide in the Discussion (p. 21-22) consideration of the individual repeats versus overall binding affinity:

“Finally, our findings indicate a balance between individual binding specificity and total binding affinity. In the evolution of PUF proteins, TRM combinations have been selected to increase or decrease individual repeat specificity while maintaining the total binding affinity needed for biological function (ref. 51-53). Structural studies have found that several PUF proteins exhibit broader specificity through the ejection of certain “undesirable” nucleotides (ref. 54). This mechanism can provide a basis for PUF recognition of degenerative binding sites and can greatly increase the number of RNA targets *in vivo*. For example, some PUF proteins (*e.g.* yeast Puf4) use their eight repeats to bind to RNA sequences with nine or ten bases by allowing one base to be turned away from the RNA binding surface (ref. 55). Occasionally, base flipping can occur to accommodate simultaneous occupancy of the binding pockets. Wang *et al.* (ref. 36) suggest that PUF proteins likely exist with greater flexibility to allow base flipping to accommodate different modes of binding to achieve overall binding affinity. With these possibilities in mind, a multi-stage process should be considered for engineering PUF domains to bind to a target RNA. Initially, PUF designs having TRM combinations in each repeat specific for the target RNA can be used in screens to recover RNA targets that bind. Then, this information can be incorporated

to determine additional design features for improved binding. The addition of machine learning to the screening results may enable features critical for PUF-RNA binding to be readily exploited.”

(h) In the discussion, for the results of this study and PUF design in general could the authors please consider how the results of Gupta *et al.* (PMID 18328718) and Wang *et al.* (PMID 19901328), where nucleotides in bound RNAs were observed to flip out of the protein binding site, might need to be considered.

We agree that the results from Gupta *et al.* (PMID 18328718) that PUF proteins can exhibit broader specificity through the ejection of an “undesirable” nucleotide and the results from Wang *et al.* (PMID 19901328) that base flipping can occur to accommodate simultaneous occupancy of several binding pockets need to be considered for future PUF designs.

We added these points to the Discussion as indicated in (g) above.

(i) The reference for Gibson assembly is an online manual from NEB (reference 30). It would be more appropriate to cite the original scientific paper: PMID 19363495.

We now cite the original scientific paper for Gibson assembly.

Reviewer #2 (Remarks to the Author):

The submitted manuscript by Zhou *et al.* describes a yeast three-hybrid approach combined with next generation sequencing to expand the RNA-binding specificity of a PUF domain, which comprises a modular architecture of eight repeats binding to an RNA 8-mer sequence. As such, it provides a bona fide target for the design of an RNA-binding domain with programmable specificity and applicability in research and therapies. The authors here combine randomizing both the PUF domains TRM in each of the eight repeats and any of the four bases for the cognate RNA sequence position.

Their data reveal not only an extended set of TRM combinations for either of the bases, but also show a repeat-specific sequence pattern. From that, Zhou *et al.* derived an extended RNA-binding code considering each repeat and base. This code has then been challenged against a set of differentially constructed target RNAs, for which cognate domain versions have been tested, incl. multiple changes within the RNA sequence with respect to the wildtype.

Altogether, I highly appreciate the systematic, initially unbiased approach underlying the non-trivial scientific challenge. The herein used Y3H system may not be able to fully cope with the complexity, but the analysis of data seems straightforward and allows for a full description of sequence preferences on both the protein and RNA sides. The findings are in principle trustworthy, convincing, the data presented in a comprehensible way, even for a non-specialist and despite the complex genetic setup. As of its current stage, the manuscript however appears to be a bit more of a resource and will require some more validation of the suggested RNA-protein code. I would thus suggest the manuscript for publication, but for this, some additional points will have to be addressed by the authors.

For a broader exploitation of findings, I still see 2-3 major necessities of additional data:

- While I do see the gain in new information from the presented data, and possible benefits from them, I am skeptical about the sole relying on the two scores for interaction and specificity. The authors should revalidate some of the major findings with experiments that allow for an alternative quantification of interactions between PUF and the RNA 8-mer. They might e.g. pick a few of their designed PUF domains, produce them recombinantly, and measure differential affinities with the cognate RNAs via techniques like SPR, ITC or similar ones, including the WT references (both protein and RNA), and control mutants. Those measurements will support the concept of real specificity outside the Y3H context and help to estimate the range of affinities in the context of the claimed expanded specificity. In light of the above-mentioned and what I did not get from the manuscript: Do the authors also find a new combination of PUF-RNA that suggests a higher affinity than the WT-WT pair? This should be used in an *in vitro* assay as well.

We agree that *in vitro* measurements help support our findings from the three-hybrid context. We initially tried the Dianthus instrument to obtain binding data but did not obtain reproducible results. We then switched to electrophoretic mobility shift assays with 20-mer RNAs.

The new results are in an additional Figure 7, in which we compare the gel assays of the wild type PUF domain binding, and of two PUF variants that each have two repeats swapped with other TRM combinations to alter the bases that are recognized. The PUF domains are tested against an RNA with the wild type sequence or with an RNA that has two bases changed to match a PUF variant. These gel assays indicate that RNA binding of the wild type and the variants is highly specific, and that the TRM combinations that are swapped into the domain result in biochemical affinities comparable to the wild type domain binding to the wild type RNA sequence. We added a new section to the manuscript, ***In vitro* binding of purified variant PUF domains**, and accompanying Methods text for this section.

We did not find a combination of a PUF variant and its target RNA with any significant increase in affinity over the wild type pair.

We attempted to purify PUF domains with three or more mutated repeats, but were not able to obtain these in soluble form. We note in the manuscript that other publications have also reported difficulties in obtaining soluble PUF domains after mutagenesis.

- Seeing the strong benefit from repeat-resolved specificity for new PUF design, what would the authors suggest as a potential design pipeline, e.g. assuming a random 8-mer to be targeted with a perfect PUF module?! How would this outdo previous rational designs that were simply based on TRM specificity per base. Could the authors use the in vitro binding assay with a suitable example RNA?!

We thank the reviewer for this interesting suggestion. We now propose at the end of the Discussion (p. 22) a possible iterative procedure to use the data we have obtained in a potential design pipeline.

I also have some minor points to be addressed:

- There is no Supplementary Figure S4 as part of the Supplement although it is referred to in the text on page 7.

Supplementary Figure S4 has been added to the manuscript

- I was not able to identify Supplemental Files 1 and 2. Not sure, but they were not accessible for me.

Supplementary File 1 and 2 are now accessible for review.

- I might have missed this, but could authors please explain the ratio for the choice of the four TRMs per base and repeat position in Figure 5?!

We are sorry that this was not clear. For Figure 5, the four TRMs were representative of TRMs showing different behavior in different repeats. The heatmap shows the normalized interaction score for each base. The color intensity represents the relative interaction score normalized by the maximal value for each row. We have changed the legend of Figure 5 to clarify this figure.

- Considering the seemingly important role of the sequence context: wouldn't one have to expect a yet bigger influence of flanking RNA sequences (beyond the 8-mer core), taking into account they might (accidentally) provoke a register shift along repeats, modulate local RNA backbone geometry, base stacking that extends into the flanking nucleotides etc. and thus influence the

precise fit of a designed PUF domain, especially in a potential application? The authors have made all their conclusions from sequences in an identical context. While this is of course necessary within the experimental logics, the effects of differential flanking sequences, e.g. comparing the 5' vs. 3' halves' robustness to changes, might not be neglectable. The authors should comment on that!

We have added new text to deal with the possibility of binding shifts, as well as an additional figure (Supplementary Figure 5). The text now includes (p. 15):

“To determine whether flanking RNA bases beyond the 8-mer core provoked a register shift along a repeat that influenced the binding of the designed PUF domains, we compared the enrichment score of the target RNA to RNAs containing possible mismatched bases (Figure S5). For example, for any design, if a 1-base 5' shift occurred in recognition, then the enrichment score of the designed 8-mer target would be similar to the three 8-mers that have the same seven 5' bases and a different final base than the target in position 8; if a 2-base shift occurred, then the enrichment score of the designed 8-mer target would be similar to the nine 8-mers that have the same six 5' bases and a different final base than the target in position 7 or 8. Similar considerations would apply at the other end of the 8-mer if 3' shifts occurred. We plotted the enrichment scores of these alternative 8-mers and found no evidence that shifting occurred for designs that bound to their target sequences; shifting may have occurred for some designs, such as designs 13, 15 and 16, that did not bind to their target sequences (Figure S5).”

Reviewer #3 (Remarks to the Author):

This manuscript describes an analysis of the RNA binding specificity of the human Pumilio-1 protein, a PUF protein that has already been the focus of previous studies. This new study is considerably more systematic and detailed than any of the preceding ones and provides a lot of data on the base preferences of the motifs at different positions in the protein. Although this is an impressively thorough and careful piece of work that will be of great help to researchers on PUF proteins, I have some doubts about the broader general interest of the study as the results tend to indicate that this may not be the best way forward if the goal is to develop 'the ability to design a protein that can bind specifically to a target RNA and regulate its fate'.

1. The focus of the manuscript is on the potential use of this data to rationally design PUF proteins to bind novel target sequences, and some successful demonstrations are included. However, the conclusion reached by readers of this work is likely to be that PUF proteins (or at least Pumilio-1) are relatively poor candidates for such rational design. This is due to the prevalence of position-specific effects (the base recognition 'code' differs at different positions, Figs 3-5) and apparent long range interactions between distant motifs (changes at one end of the protein affect binding at the other end, Fig 6 and Discussion). The result is that only relatively

minor differences to the target sequence can be tolerated whilst still achieving strong and predictable binding. The ease of design and the predictability of binding is certainly lower than for Cas13 and related RNA-guided proteins, and even appears to be worse than for PPR proteins, which can be relatively easily designed to bind novel targets (e.g. Miranda et al *Nucleic Acids Research*, 46: 2613–2623; Colas des Francs-Small et al. *Communications Biology*, 1: 166; Yan et al. *Nucleic Acids Research*, 47: 3728–3738; McDermott et al. *Plant Cell*, 31:1723–1733). Cas13 and PPR proteins are only mentioned in passing (and summarily dismissed) in the introduction; a discussion of their relative merits would be appropriate in the discussion.

As we indicate in response to similar comments from Reviewer #2, we have provided a more thorough and, we hope, balanced discussion of other RNA-binding approaches in the Introduction. In addition, we have cited the relevant publications that deal with the other approaches.

2. I suspect that some (many?) of the complexities encountered in this work are due to the decision to start with a natural PUF protein (human Pumilio-1) rather than a synthetic protein based on quasi-identical consensus motifs. The latter approach has proved to be a very powerful means to simplify and improve rational design of both PUF and PPR proteins. It seems likely that this would reduce position-specific effects (and, less likely, the long range interactions). I would like to see a justification for the approach that was chosen. A potential advantage to the use of a natural protein is the ability to make use of millions of years of evolution in the exploration of the sequence space. This advantage is not used here. A discussion of sequence variability in Pumilio-1 homologues and their target sites might nicely complement the experimental results.

We chose to use a natural PUF protein (human Pumilio-1) rather than a synthetic protein in order to compare the high-throughput data we obtained with the TRM combinations that occur in nature.

Below is a figure from Campbell et al. 2014 that scored the prevalence of TRM combinations in Pumilio-1 homologues.

We now include in the Discussion (p. 20):

“Campbell *et al.* (ref. 28) scored the prevalence of TRM combinations at each PUF repeat in 94 Pumilio-1 homologues, inferring the abundance of natural TRM combinations from the sequence alignments. Their aligned data are broadly consistent with our high-throughput screening results. It is striking that the C-terminal repeats of the PUF domain can be rationally designed to bind other RNA sequences, yet are highly conserved in their specificity during evolution. As has been noted (ref. 28), the observation that a wide range of PUF proteins maintain similar C-terminal TRMs and a UGU sequence at the 5' end of the binding site implies that this region executes biological roles beyond RNA binding that constrain the protein's evolutionary divergence.”

4. Concerning the data itself, everything in manuscript comes from a rather indirect assay, namely a high-throughput three-hybrid screen in yeast cells. Although the data look good and I see no particular issues with these experiments, I would have liked to see some confirmation by more direct biochemical means (e.g. *in vitro* measurements of binding affinity and specificity by one of ITC, MST, FP, SPR, REMSA...)

Please see response to Reviewer #2 comments about our *in vitro* experiments.

5. The explanations proposed for position-specific effects appear to focus on possible interactions within the PUF motifs (or between neighboring motifs). I wonder if the authors have looked at possible nearest-neighbor effects in the RNA? As stacking interactions are clearly significant for binding and specificity, the nature of the neighboring 3' nucleotide (especially whether it is a purine or pyrimidine) might need to be taken into account. Currently, the fact that the amino acid sidechain at position 13 contacts two bases doesn't appear to be considered. Even if it can be easily eliminated as a factor it would be worth discussing.

We agree that the possibility of the amino acid sidechain at position 13 may contact two bases could contribute to the position-specific effects. We now consider this nearest-neighbor effect in the Discussion (p. 21):

“Koh *et al.* (ref 51) tested whether the identity of the stacking residue contributes to specificity for the neighboring 3' base and found that the effect is limited. Our data suggest a similar conclusion. For example, with mutation of SNE to NPG in repeat 7, the substitution from asparagine to proline in the stacking residue did not alter the preference for U at the neighboring 3' base (base 3) (Figure 6). The same phenomenon can be seen in other designs as well, which indicate a limited nearest-neighbor effect resulting from the stacking residue substitution.”

6. When the natural site is modified, can we be sure that the protein variants will bind to exactly the same position? All the analyses are assuming no shift in binding, but it's not clear how safe this assumption is.

Please see the response to Reviewer #2 comments about possible binding shifts.

REVIEWERS' COMMENTS

Reviewer #1 (Remarks to the Author):

In this revision Zhou and colleagues have comprehensively addressed the points raised in the initial round of review. The more detailed definition of the RNA recognition code for the PUM1 PUF protein that they provide should be of great interest to the field and I look forward to seeing the published manuscript.

Reviewer #2 (Remarks to the Author):

The authors have addressed my concerns and I recommend the manuscript for publication.

Reviewer #3 (Remarks to the Author):

The authors have replied in depth to each of the questions and comments and I believe that all of the potential weaknesses of the initial version have been adequately dealt with.